# FROM ADAPTIVE QUERY RELEASE TO MACHINE UNLEARNING

## ABSTRACT

We formalize the problem of machine unlearning as design of efficient unlearning algorithms corresponding to learning algorithms which perform a selection of adaptive queries from structured query classes. We give efficient unlearning algorithms for linear and prefix-sum query classes. As applications, we show that unlearning in many problems, in particular, stochastic convex optimization (SCO), can be reduced to the above, yielding improved guarantees for the problem. In particular, for smooth Lipschitz losses and any $\rho > 0$, our results yield an unlearning algorithm with excess population risk of $\widetilde{O}\left(\frac{1}{\sqrt{n}} + \frac{\sqrt{d}}{n\rho}\right)$ with unlearning query (gradient) complexity $\widetilde{O}(\rho \cdot \text{Retraining Complexity})$, where $d$ is the model dimensionality and $n$ is the initial number of samples. For non-smooth Lipschitz losses, we give an unlearning algorithm with excess population risk $\widetilde{O}\left(\frac{1}{\sqrt{n}} + \left(\frac{\sqrt{d}}{n\rho}\right)^{1/2}\right)$ with the same unlearning query (gradient) complexity. Furthermore, in the special case of Generalized Linear Models (GLMs), such as those in linear and logistic regression, we get dimension-independent rates of $\widetilde{O}\left(\frac{1}{\sqrt{n}} + \frac{1}{(n\rho)^{2/3}}\right)$ and $\widetilde{O}\left(\frac{1}{\sqrt{n}} + \frac{1}{(n\rho)^{1/3}}\right)$ for smooth Lipschitz and non-smooth Lipschitz losses respectively. Finally, we give generalizations of the above from one unlearning request to *dynamic* streams consisting of insertions and deletions.

## 1 INTRODUCTION

The problem of machine unlearning is concerned with updating trained machine learning models upon request of deletions to the training dataset. This problem has recently gained attention owing to various data privacy laws such as General Data Protection Regulation (GDPR), California Consumer Act (CCA) among others, which empower users to make such requests to the entity possessing user data. The entity is then required to update the state of the system such that it is *indistinguishable* to the state had the user data been absent to begin with. While as of now, there is no universally accepted definition of *indistinguishibility* as the unlearning criterion, in this work, we consider the most strict definition, called **exact unlearning** (see Definition 1 for a formal definition).

**Motivating Example:** The main objective of our work is to identify algorithmic design principles for unlearning such that it is more *efficient* than retraining, the naive baseline method. Towards this, we first discuss the example of unlearning for Gradient Descent (GD) method, which will highlight the key challenges as well as foreshadow the formal setup and techniques. GD and its variants are extremely popular optimization methods with numerous applications in machine learning and beyond. In a machine learning context, it is typically used to minimize the training loss, $\widehat{L}(w; S) = \frac{1}{n} \sum_{i=1}^{n} \ell(w; z_i)$ where $S = \{z_i\}_{i=1}^{n}$ is the training dataset and $w$, the model. Starting from an initial model $w_1$, in each iteration, the model is updated as:

$$w_{t+1} = w_t - \eta \nabla \widehat{L}(w_t; S) = w_t - \eta \left( \frac{1}{n} \sum_{i=1}^{n} \nabla \ell(w_t; z_i) \right).$$

After training, a data-point, say $z_n$ without loss of generality, is requested to be unlearnt and so the updated training set is $S' = \{z_i\}_{i=1}^{n-1}$. We now need to apply an *efficient* unlearning algorithm such that its output is equal to that of running GD on $S'$. Observe that the first iteration of GD is *simple* enough to be unlearnt efficiently by computing the *new* gradient $\nabla \widehat{L}(w_0; S') = \frac{1}{n-1}\left(n \nabla \widehat{L}(w_1; S) - \nabla \ell(w_1; z_n)\right)$ and updating as $w_2' = w_1 - \eta \nabla \widehat{L}(w_1; S')$. However, in the second iteration (and

onwards), the gradient is computed on $w_2'$ which can be different from $w_2$ and the above adjustment can no longer be applied and one may need to retrain from here onwards. This captures the key challenge for unlearning in problems solved by *simple* iterative procedures such as GD – **adaptivity** – i.e., the gradients (or more generally, the *queries*) computed in later iteration depend on the result of the previous iterations. We systematically formalize such procedures and design efficient unlearning algorithms for them. We summarize our key contributions below.

## 1.1 OUR RESULTS AND TECHNIQUES

**Learning/Unlearning as Query Release:** Iterative procedures are an integral constituent of the algorithmic toolkit for solving machine learning problems and beyond. As in the case of GD above, these often consist of a sequence of *simple* but *adaptive* computations. The simple computations are often efficiently undo-able (as in the first iteration of GD) but its *adaptive* nature – change of result of one iteration changing the trajectory of the algorithm – makes it difficult to undo computation, or unlearn, efficiently.

As opposed to designing unlearning (and learning) algorithms for specific (machine learning) problems, we study the design of unlearning algorithms corresponding to (a class of) learning algorithms. We formalize this by considering learning algorithms which perform *adaptive query release* on datasets. Specifically, this consists of a selection of adaptive queries from structured classes like linear and prefix-sum queries (see Section 3 for details). The above example of GD is an instance of linear query, since the query, which is the average gradient $\frac{1}{n}\sum_{i=1}^{n}\nabla\ell(w_t; z_i)$, is linear in the data-points. With this view, we study how to design *efficient* unlearning algorithms for such methods.

We use efficiency in the sense of number of queries made (query complexity), ignoring the use of other resources, e.g., space, computation for selection of queries, etc. To elaborate on why this is interesting, firstly note that this does not make the problem trivial, in the sense that even with unlimited access to other resources, it is still challenging do design an unlearning algorithm with query complexity smaller than that of retraining (the naive baseline). Secondly, let us revisit the motivation from solving optimization problems. The standard model to measure computation in optimization is the number of gradient queries a method makes for a target accuracy, often abstracted in an oracle-based setup (Nemirovskij & Yudin, 1983). Importantly, this setup imposes no constraints on other resources, yet it witnesses the optimality of well-known simple procedures like (variants of) GD. We follow this paradigm, and as applications of our results to stochastic convex optimization (SCO), we make progress on the fundamental question of understanding the gradient complexity of unlearning in SCO. Interestingly, our proposed unlearning procedures are simple enough that the improvement over retraining in terms of query complexity also applies even with accounting for the (arithmetic) complexity of all other operations in the learning and unlearning methods.

**Linear queries:** The simplest query class we consider is that of linear queries (details deferred to Appendix B). Herein, we show that the prior work of Ullah et al. (2021), which focused on unlearning in SCO and was limited to the stochastic gradient method, can be easily extended to general linear queries. This simple observation yields unlearning algorithms for algorithms for federated optimization, $k$-means clustering, etc. Herein, we give a $\rho$-TV stable (see Definition 2) learning procedure with $T$ adaptive queries and a corresponding unlearning procedure with a $O(\sqrt{T}\rho)$ relative unlearning complexity (the ratio of unlearning and retraining complexity; see Definition 4).

**Prefix-sum queries:** Our main contribution is the case when we consider the class of prefix-sum queries. These are a sub-class of interval queries which have been extensively studied in differential privacy and are classically solved by the binary tree mechanism (Dwork et al., 2010). We note in passing that for differential privacy, the purpose of the tree is to enable a tight privacy accounting and no explicit tree may be maintained. In contrast, for unlearning, we show that maintaining the binary tree data structure aids for efficient unlearning. We give a binary-tree based $\rho$-TV stable learning procedure and a corresponding unlearning procedure with a $\widetilde{O}(\rho)$ relative unlearning complexity.

**Unlearning in Stochastic Convex Optimization (SCO):** Our primary motivation for considering prefix-sum queries is its application to unlearning in SCO (see Section 2 for preliminaries).

**1. Smooth SCO**: The problem of unlearning in smooth SCO was studied in Ullah et al. (2021) which proposed algorithms with excess population risk of $O\left(\frac{1}{\sqrt{n}} + \left(\frac{\sqrt{d}}{n\rho}\right)^{2/3}\right)$ where $\rho$ is the relative

| Problem | Base algorithm | Rate |
|---|---|---|
| Smooth, Lipschitz-SCO | Variance Reduced-Frank Wolfe | $\frac{1}{\sqrt{n}} + \frac{\sqrt{d}}{n\rho}$ |
| Lipschitz SCO | Dual Averaging | $\frac{1}{\sqrt{n}} + \frac{d^{1/4}}{\sqrt{n\rho}}$ |
| Smooth, Lipschitz GLM | JL + Variance Reduced-Frank Wolfe | $\frac{1}{\sqrt{n}} + \frac{1}{(n\rho)^{2/3}}$ |
| Lipschitz GLM | JL + Dual Averaging | $\frac{1}{\sqrt{n}} + \frac{1}{(n\rho)^{1/3}}$ |

Table 1: Excess population risk guarantees for various problems as well as the base algorithm; $\rho$: relative unlearning complexity (see Definition 4), JL: Johnson-Lindenstrauss transform.

unlearning complexity. We show that using a variant of variance-reduced Frank-Wolfe (Zhang et al., 2020), which uses prefix-sum queries, yields an improved excess population risk of $O\left(\frac{1}{\sqrt{n}} + \frac{\sqrt{d}}{n\rho}\right)$.

**2. Non-smooth SCO**: In the non-smooth setting, both algorithms proposed in Ullah et al. (2021) have trivial guarantees. Instead we give an algorithm based on Dual Averaging (Nesterov, 2009), which again uses prefix-sum query access, and thus fits into the framework. This algorithm gives us an excess population risk of $O\left(\frac{1}{\sqrt{n}} + \frac{d^{1/4}}{\sqrt{n\rho}}\right)$.

**3. Generalized Linear Models (GLM)**: Generalized linear models are one of most basic machine learning problems which include the squared loss (in linear regression), logistic loss (in logistic regression), hinge loss (support vector machines) etc. We study unlearning in two classes of GLMs (see below), for which we combine recently proposed techniques based on dimensionality reduction (Arora et al., 2022) with the above prefix-sum query algorithms to get the following dimension-independent rates.

**3(a). Smooth GLM:** For the smooth GLM setting, we combine Johnson-Lindenstrauss transform with variance reduced Frank-Wolfe to get $O\left(\frac{1}{\sqrt{n}} + \frac{1}{(n\rho)^{2/3}}\right)$ excess population risk. Note that we get no overhead in statistical rate even with very small relative unlearning complexity, $\rho \approx n^{-1/4}$. This class of smooth GLMs contains the well-studied problem of logistic regression. Hence, our result demonstrates that it is possible to unlearn logistic regression with sub-linear (specifically, $O(n^{3/4})$) unlearning complexity with no sacrifice in the statistical rate.

**3(b). Lipschitz GLM:** Similarly, for the Lipschitz convex GLM setting, we combine Johnson-Lindenstrauss transform with dual averaging yielding a rate of $\widetilde{O}\left(\frac{1}{\sqrt{n}} + \frac{1}{(n\rho)^{1/3}}\right)$.

Please see Table 1 for a summary of above results.

**SCO in dynamic streams:** Finally, we consider SCO in dynamic streams (details deferred to Appendix F) where we observe a sequence of insertions and deletions and are supposed to produce outputs after each time-point. In this case, we present two methods: one which satisfies the exact unlearning guarantee with worse update time, the other which satisfies weak unlearning (see Definition 9) with improved update time. The exact unlearning method is inspired from the work of Ullah et al. (2021) which dealt with insertions similar to deletions. The weak unlearning method is motivated from the observation that the above may be too pessimistic. To elaborate, inserting a new data item does not warrant a (unlearning) guarantee that the algorithm's state be indistinguishable to the case if the point was not inserted. Hence, insertions should require smaller (ideally, constant) update time which is indeed the case for our proposed methods.

## 1.2 RELATED WORK

Our work is a direct follow up of Ullah et al. (2021) which proposed the framework of Total Variation (TV) stability and maximal coupling for the **exact** machine unlearning problem. They applied this to unlearning in smooth stochastic convex optimization (SCO) and obtained a guarantee of $\frac{1}{\sqrt{n}} + \left(\frac{\sqrt{d}}{n\rho}\right)^{\frac{2}{3}}$ on excess population risk, where $n$ is the number of data samples, $d$, model dimensionality and $\rho$ is the relative unlearning complexity (see Definition 4). We improve upon the results in that work in multiple ways as described in the preceding section. Furthermore, the exact unlearning problem has been studied for $k$-means clustering (Ginart et al., 2019) and random forests (Brophy & Lowd, 2021). The work of Bourtoule et al. (2021) proposes a general methodology for exact unlearning for deep learning methods but do not provide rigorous theoretical guarantees on accuracy, even in simple settings. Besides this, there are works which consider unlearning in SCO, however they use an *approximate* notion of unlearning inspired from differential privacy (Guo et al., 2019; Neel et al., 2021; Sekhari et al., 2021; Gupta et al., 2021), and therefore are incomparable to our work.

## 2 SETUP AND PRELIMINARIES

Let $\mathcal{Z}$ be the data space, $\mathcal{W}$ be the model space and $\mathcal{M}$ be the *meta-data* space, where meta-data is additional information a learning algorithm may save to aid unlearning. We consider a learning algorithm as a map $\mathbf{A} : \mathcal{Z}^* \to \mathcal{W} \times \mathcal{M}$ and an unlearning algorithm as a map $\mathbf{U} : \mathcal{W} \times \mathcal{M} \times \mathcal{Z} \to \mathcal{W} \times \mathcal{M}$. We use $\mathcal{A}$ and $\mathcal{U}$ to denote the first output (which belongs to $\mathcal{W}$) of $\mathbf{A}$ and $\mathbf{U}$ respectively.

We recall the definition of exact unlearning from Ullah et al. (2021) which requires that the entire state after unlearning be indistinguishable from the state obtained if the learning algorithm were applied to the dataset without the deleted point.

**Definition 1** (Exact unlearning). *A procedure* $(\mathbf{A}, \mathbf{U})$ *satisfies exact unlearning if for all datasets $S$, all $z \in \mathcal{Z}$, and for all events $\mathcal{E} \subseteq \mathcal{W} \times \mathcal{M}$, we have,* $\mathbb{P}\left(\mathbf{A}\left(S\setminus\{z\}\right) \in \mathcal{E}\right) = \mathbb{P}\left(\mathbf{U}\left(\mathbf{A}(S), z\right) \in \mathcal{E}\right)$

**Unlearning request:** We consider the setting where we start with a dataset of $n$ samples and observe **one** unlearning request. We assume that the choice of unlearning request is oblivious to the learning process. In Section F, we generalize our result to the streaming setting.

**Total Variation stability, maximal coupling and efficient unlearning:** The Total Variation (TV) distance between two probability distributions $P$ and $Q$ is $\mathsf{TV}(P, Q) = \sup_{\text{measureable } \mathcal{E}} |P(\mathcal{E}) - Q(\mathcal{E})|$. Next, we define Total Variation (TV) stability, which was introduced in Ullah et al. (2021) to motivate algorithmic techniques for efficient unlearning.

**Definition 2.** *An algorithm $\mathcal{A}$ is said to be $\rho$ Total Variation stable if for all datasets $S$ and $S'$ differing in one point, i.e.* $|S\Delta S'| = 1$*, the total variation distance,* $\mathsf{TV}\left(\mathcal{A}(S), \mathcal{A}(S')\right) \leq \rho$

Given two distributions $P$ and $Q$, a **coupling** is a joint distribution $\pi$ with marginals $P$ and $Q$. Furthermore, a **maximal coupling** is a coupling $\pi$ such that the disagreement probability $\mathbb{P}_{(x,y)\sim\pi}\{x \neq y\} = \mathsf{TV}(P, Q)$. In the unlearning context, $P = \mathcal{A}(S)$, the output on initial dataset, and $Q = \mathcal{A}(S')$, the output on the updated dataset. Hence, the unlearning problem simply becomes about transporting $P$ to $Q$ with small *computational cost*, akin to optimal transport (Villani, 2009). Furthermore, when sampled from a maximal coupling between $P$ and $Q$, by definition, we get the **same sample** for both $P$ and $Q$, expect with probability $\rho$, and yet satisfying the exact unlearning criterion. The main idea is that for certain learning algorithms of interest, during unlearning, we can **efficiently** construct a (near) maximal coupling of $P$ and $Q$, and so the same model output from $P$ suffices for $Q$, most of the times. In particular, the fraction of times that we need change the model is the TV-stability parameter $\rho$ of the learning algorithm. The goal, therefore, is to design an (accurate) TV-stable learning algorithm and a corresponding efficient coupling-based unlearning algorithm. In this work, we use the technique of reflection coupling described below.

**Reflection Coupling (Lindvall & Rogers, 1986):** Reflection Coupling is a classical technique in probability to maximally couple symmetric probability distributions. Consider two probability distributions $P$ and $Q$ with means $u$ and $u'$ and let $r$ be a sample from $P$. The process involves a rejection sampling step on the two distributions and sample $r$ (see line 13 in in Algorithm 3). If it results in accept, we use the same $r$ as the sample from $Q$, otherwise, we apply the following map: $\text{Reflect}(u, u', r) = u - u' + r$, which gives the sample from $Q$ (see line 16 in Algorithm 3).

Our algorithmic techniques borrow tools from differential privacy (Dwork et al., 2014) such as its relationship with Total Variation stability; we describe these in Appendix A.

**Stochastic Convex Optimization (SCO):** SCO is the dominant framework for computationally-efficient machine learning. Consider a convex (constraint) set $\mathcal{W}$ and let $D$ denote its diameter. Let $\ell : \mathcal{W} \times \mathcal{Z} \to \mathbb{R}$ be a loss function, which is convex in its first parameter $\forall z \in \mathcal{Z}$. Given $n$ i.i.d. points from an unknown probability distribution $\mathcal{D}$ over $\mathcal{Z}$, the goal is to devise an algorithm, the output of which has small population risk, defined as $L(w; \mathcal{D}) := \mathbb{E}_{z \in \mathcal{D}} \ell(w; z)$. The excess population risk is then $L(w; \mathcal{D}) - L(w^*; \mathcal{D})$ where $w^*$ denotes a population risk minimizer over $\mathcal{W}$.

**Generalized Linear Models (GLM):** Generalized Linear Models (GLMs) are loss functions typically encountered in supervised learning problems, like linear and logistic regression. Herein, $\ell(w; (x, y)) = \phi_y(\langle w, x \rangle)$, where $\phi_y : \mathbb{R} \to \mathbb{R}$ is some function. We use $\|\mathcal{X}\|$ to denote the radius bound on data points, i.e. for $x \in \mathcal{X} \subseteq \mathbb{R}^d$, $\|x\| \leq \|\mathcal{X}\|$. In this case, we consider the unconstrained setup i.e. $\mathcal{W} = \mathbb{R}^d$, as it allows to get dimension-independent rates for GLMs, similar to what happens under differential privacy (Jain & Thakurta, 2014; Arora et al., 2022).

---

**Algorithm 1** Template learning algorithm

---

**Input:** Dataset $S$, steps $T$, query functions $\{q_t(\cdot)\}_{t \leq T}$ where $q_t \in \mathcal{Q}$, a query class, update functions $\{U_t(\cdot)\}_{t \leq T}$, selector function $\mathcal{S}(\cdot)$

1: Initialize model $w_1 \in \mathcal{W}$
2: **for** $t = 1$ to $T - 1$ **do**
3:     Query dataset $u_t = q_t \left( \{w_i\}_{i \leq t}, S \right)$
4:     Update $w_{t+1} = U_t(\{w_i\}_{i \leq t}, u_t)$
5: **end for**
**Output:** $\widehat{w} = \mathcal{S} \left( \{w_t\}_{t \leq T} \right)$

---

We introduce the Johnson-Lindenstrauss property below which is crucial to our construction.

**Definition 3** (Johnson-Lindenstrauss property). *A random matrix $\Phi \in \mathbb{R}^{k \times d}$ satisfies $(\beta, \gamma)$-JL property if for any $u, v \in \mathbb{R}^d$, with probability at least $1 - \gamma$, $\mathbb{P}\left(|\langle \Phi u, \Phi v \rangle - \langle u, v \rangle| \geq \beta \|u\| \|v\|\right) \leq \gamma$.*

There exists many efficient constructions of such random matrices (Nelson, 2011).

## 3    UNLEARNING FOR ADAPTIVE QUERY RELEASE

We now set up the framework of adaptive query release, which is a lens to view (existing) iterative learning procedures; this view is useful in our design of corresponding unlearning algorithms. Iterative procedures run on datasets consist of a sequence of *interactions* with the dataset; each interaction computes a certain function, or query, on the dataset. The chosen query is typically adaptive, i.e., dependent on the prior query outputs. We consider iterative learning procedures which are composed of adaptive queries from a specified query class. Formally, consider a query class $\mathcal{Q} \subseteq \mathcal{W}^{\mathcal{W}^* \times \mathcal{Z}^*}$; herein, each query in $\mathcal{Q}$ is a function of a sequence of $\{w_i\}_{i < t}$ (typically, prior query outputs), and the dataset $S$, with output in $\mathcal{W}$. With this view, we give a general template of a learning procedure as Algorithm 1, where $\{U_t\}_t$ and $\mathcal{S}$ are the update and selector functions internal to the algorithm.

**Query Model:** We describe the query model which we use to measure computational complexity. Under the model, a query function $q(\{w\}_i, S)$ takes $|S|$ unit computations (or queries, for brevity) for any $q$ and $\{w_i\}_i$. In our applications to SCO, this will correspond to the gradient oracle complexity.

Our algorithmic approach to unlearning is rooted in the relationship between TV stability and maximal couplings. With this view, for a specified query class, we have the following requirements.

1. **TV-stability:** We want a $\rho$-TV stable "modification" (clarified later) of the learning Algorithm 1.

2. **Efficient unlearning algorithm**: We measure efficiency as the average number of queries the unlearning algorithm makes relative to the learning algorithm (retraining), defined as follows.

   **Definition 4** (Relative Unlearning Complexity). *The Relative Unlearning Complexity is defined as*
   $$\frac{\mathbb{E}_{(\mathbf{A}, \mathbf{U})} \left[\text{Query complexity of unlearning algorithm } \mathbf{U}\right]}{\mathbb{E}_{\mathbf{A}} \left[\text{Query complexity of learning algorithm } \mathbf{A}\right]}$$

   For a $\rho$-TV stable learning algorithm, we want that the relative unlearning complexity is (close to) $\rho$. This is because our proposed unlearning algorithm constructs a (near) maximal coupling of the learning algorithm's output under the original and updated dataset. This means that unlearning algorithm changes the original output (under the original dataset) with probability at most $\rho$ – we want that in this case, the unlearning algorithm makes a number of queries akin to retraining. We also want that in the other case when it does change the output, it makes a small (ideally, constant) number of queries. We note that relative unlearning complexity, in itself, does not completely capture if the unlearning algorithm is *good*, since it may be the case that the corresponding learning algorithm is computationally more expensive than other existing methods. However, in our applications to SCO (Section 5), all the learning algorithms are linear time, so the denominator in the definition above is as small as it can be (asymptotically), i.e. $\Theta(n)$.

3. **Accuracy:** We will primarily be concerned with correctness of the unlearning algorithm and its efficiency. In the applications (Section 5), we will give accuracy guarantees for specific problems, where we will see (roughly) that if the learning algorithm is *noise-tolerant*, then the solution produced by the proposed TV stable modified algorithm are still accurate.

## 4 PREFIX SUM QUERIES

We now consider prefix-sum queries, which is the main contribution of this work. The reason for considering this query class is that two powerful algorithms for SCO, dual averaging and recursive variance reduction fit into this template. We start by defining a prefix-sum query.

**Definition 5.** *A set of queries $\{q_t\}_{t \geq 1}$ where $q_t : \mathcal{W}^t \times \mathcal{Z}^n \to \mathcal{W}$ are called prefix-sum queries if $q_1(w_1, S) = p_1(w_1, z_1)$ and for all $t > 1$, $q_t(\{w_i\}_{i \leq t}, S) = q_{t-1}(\{w_i\}_{i < t}, S) + p_t(\{w_i\}_{i \leq t}, z_t))$ for some functions $\{p_t\}_{t \geq 1}$ where $p_t : \mathcal{W}^* \times \mathcal{Z} \to \mathcal{W}$.*

Simply put, prefix-sum queries, sequentially query **new** data points and adds them to the previous accumulated query. A simple example is computing partial sums of data points $(z_1, z_1 + z_2, \ldots)$. Note that in the above definition, we can equivalently represent the prefix-sum queries using the sequence $\{p_t\}_t$. We also assume that the queries have bounded sensitivity, defined as follows.

**Definition 6.** *A query $q : \mathcal{W}^* \times \mathcal{Z}^n \to \mathcal{W}$ is B-sensitive if*
$$\sup_{\{w_i\}_i} \sup_{S, S' : |S \Delta S'| = 1} \|q(\{w_i\}_i, S) - q(\{w_i\}_i, S')\| \leq B.$$

We note that the bounded sensitivity condition is satisfied in a variety of applications; see Section 5.

### 4.1 LEARNING WITH BINARY TREE DATA-STRUCTURE:

The learning algorithm, given as Algorithm 2, is based on answering the adaptive prefix-sum queries with the binary tree mechanism (Dwork et al., 2010). For $n$ samples (assume $n$ is a power of two, otherwise we can append dummy "zero" samples without any change in asymptotic complexity), the binary tree mechanism constructs a complete binary tree $\mathcal{T}$ with the leaf nodes corresponding to the data samples. The key idea in the binary tree mechanism is that instead of adding *fresh* independent noise to each prefix-sum query, it is *better* to add correlated noise, where the correlation structure is described by a binary tree. For example, suppose we want to release the **seventh** prefix-sum query, $\sum_{i=1}^{7} p_i(\{w_j\}_{j \leq i}, z_i)$, then consider the dyadic decomposition of 7 as $4, 2$ and $1$, and release the sum $\left(\sum_{i=1}^{4} p_i(\{w_j\}_{j \leq i}, z_i) + \xi_1\right) + \left(\sum_{i=5}^{6} p_i(\{w_j\}_{j \leq i}, z_i) + \xi_2\right) + \left(p_7(\{w_j\}_{j \leq i}, z_i) + \xi_3\right)$ where $\xi_i$'s denote the added noise, which may have also been used in prior prefix-sum query responses. See Figure 1 for a simplified description of the process.

We index the nodes of the tree using using binary numbers $B = \{0, 1\}^{\log(n)}$ which describes the path from the root. Let the tree $\mathcal{T} = \{v_b\}_{b \in B}$ which denotes the contents stored by the learning algorithm. Herein, each node contains the tuple $(u, r, w, z)$ where $u \in \mathbb{R}^d$ is the query response, $r \in \mathbb{R}^d$ is the *noisy* response, $w \in \mathbb{R}^d$ a model and $z \in \mathcal{Z}$ a data point. In fact, only the leaf nodes store the model and data sample. Finally, define $\mathsf{leaf} : [n] \to \{0, 1\}^{\log(n)}$ which gives the binary representation of the input leaf node.

This binary tree data structure supports the following operations:
1. $\mathsf{Append}(u, \sigma; \mathcal{T})$: Add a query response $u$, perturbed by noise of variance $\sigma^2$, to $\mathcal{T}$, which involves adding noisy $u$ to $u_v$ for $v$ in the path from leaf to root.

2. $\mathsf{GetPrefixSum}(t; \mathcal{T})$, where $t \in \mathbb{N}$: Get the $t$-th noisy response from $\mathcal{T}$.

3. $\mathsf{Get}(b; \mathcal{T})$ where $b \in \{0, 1\}^{\log(n)}$: Get all items in the vertex of $\mathcal{T}$ indexed by $b$.

4. $\mathsf{Set}(b, v; \mathcal{T})$ where $b \in \{0, 1\}^{\log(n)}$: Set the contents of vertex $b$ in the $\mathcal{T}$ as $v$.

Following Guha Thakurta & Smith (2013), we give pseudo-codes of the above operations in Section C, with minor modifications to aid the unlearning process.

### 4.2 UNLEARNING BY MAXIMALLY COUPLING BINARY TREES

The unlearning Algorithm 3 is based on constructing a (near) maximal coupling of the binary trees under current and updated dataset. Let $z_j$ be the element to be deleted and let $v_s$ be the leaf node which contains $z_j$ (we use $z$ in place of $z_j$ from here on, for simplicity). During unlearning, we simulate (roughly speaking) the dynamics of the learning algorithm if the deleted point was not present to begin with. In that case, in place of the deleted point, some other point would have been used. Now, since the dataset was randomly permuted, every point is equally likely to have been used, and thus we can use the point $z'$ in the last leaf node, say $v_l$, in the tree – this choice of the last point is important for unlearning efficiency. Firstly, the computations associated with the last point $z'$ needs

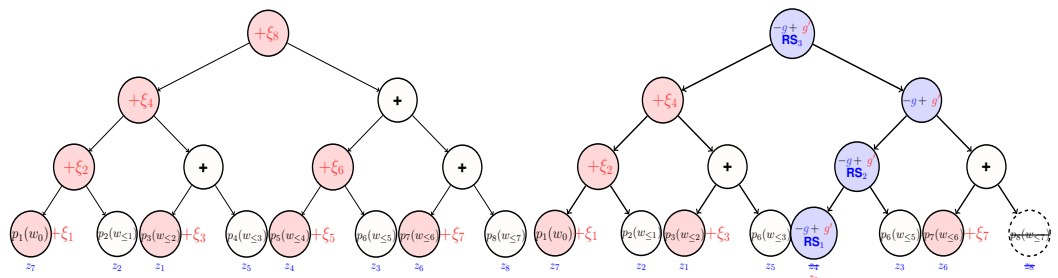

Figure 1: A simplified schematic of the learning (left) and unlearning (right) procedures for prefix-sum queries. In the left, the leaves contain (noisy, if $+\xi_i$) prefix-sum queries applied on the randomly permuted data-point ($z_i$'s) below it. The intermediate nodes with **+** adds the not-noised values of its children, where as others add noise to it. On the right, the deleted point $z_4$ is replaced with $z_8$ which amounts to adjusting the queries with $-g + g'$ (see Algorithm 3 for details) and performing Rejection Sampling (abbreviated $\mathbf{RS}_i$, where $_i$ indicates the order of occurrence) along the height of the tree.

---

**Algorithm 2** TreeLearn$(t_0; \mathcal{T})$

---

**Input:** Dataset $S$, steps $T$, $B$-sensitive prefix queries $\{p_t\}_{t \leq T}$, update functions $\{U_t\}_{t \leq T}$, $\sigma$

1: **if** $t_0 = 1$ **then** Permute dataset and initialize $\mathcal{T}$ **end if**
2: $(\cdot, \cdot, w_{t_0}, \cdot) = \mathsf{Get}(\mathrm{leaf}(t_0); \mathcal{T})$
3: **for** $t = t_0$ to $|S| - 1$ **do**
4: $\quad u_t = p_t(\{w_t\}_{i \leq t}, z_t)$
5: $\quad \mathsf{Append}(u_t, \sigma; \mathcal{T})$
6: $\quad r_t = \mathsf{GetPrefixSum}(t; \mathcal{T})$
7: $\quad w_{t+1} = U_t\left(\{w_t\}_{\leq t}, r_t\right)$
8: $\quad \mathsf{Set}(\mathrm{leaf}(t), (u_t, r_t, w_t, z_t); \mathcal{T})$
9: **end for**
**Output:** $\widehat{w} = \mathcal{S}(\{w_t\})$

---

to be undone – towards this, we update the contents of the nodes in the path from node $v_l$ to root (line 5), finally removing node $v_l$ from the tree (line 6). Then, we need to *replace* all the computations which used the deleted point $z$ with the same computation under $z'$. Since the learning algorithm was based on the binary tree mechanism, the point $z$ was only **explicitly** used in the nodes from leaf $v_s$ to the root (so, at most $\log(n)$ nodes). We say explicitly above because due to the adaptive nature of the process, in principle, all nodes after $v_s$ depend on it, in the sense that their contents would change if the response in $v_s$ were to change. However, importantly, the binary search structure of our learning algorithm and our coupling technique (details below) would enable us to (mostly) only care about explicit computations.

We first compute **two** new queries, under the data point $z$ and $z'$, with responses $g = p_j(\{w_q\}_{q \leq s}, z)$ and $g' = p_j(\{w_q\}_{q \leq s}, z')$ respectively (line 3). Starting with leaf node $v_s$, we update the original unperturbed prefix-sum query response under $z$ i.e. $u$ to what it would have been under data-point $z'$: $u' = u - g' + g$ (line 11). Further, since the training method adds noise $\mathcal{N}(0, \sigma^2 \mathbb{I})$ to $u$ to produce original noisy response $r$, we now need to produce a sample from $\mathcal{N}(u', \sigma^2 \mathbb{I})$ to satisfy exact unlearning. Naively, we could simply get a *fresh* independent sample from $\mathcal{N}(u', \sigma^2 \mathbb{I})$, however, this would change the noisy response $r$, and hence require all subsequent computations to be redone (the adaptive nature). So, ideally, we want to reuse the same $r$ and yet generate a sample from $\mathcal{N}(u', \sigma^2 \mathbb{I})$. This is precisely the problem of constructing a maximal coupling, discussed in the Section 2, wherein we also discussed the method of reflection coupling to do it. This amounts to doing a rejection sampling which (roughly) ascertains if response $r$ is still sufficient under the new distribution $\mathcal{N}(u', \sigma^2 \mathbb{I})$. Specifically we compute the ratio of the probability densities at $r$ under the noise added to $u$ and $u'$, i.e. $\frac{\phi_{\mathcal{N}(u, \sigma^2 \mathbb{I})}(r)}{\phi_{\mathcal{N}(u', \sigma^2 \mathbb{I})}(r)}$ and compare it against a randomly sampled $\mathrm{Unif}(0,1)$; if it results in accept, we move to parent of the node $v_s$, and repeat. If any step fails, we reflect which generates a different noisy response $r'$, and continue retraining from the next leaf w.r.t. the post order traversal of the tree (the variable ct in Algorithm 3 keeps track of this *next* node). See Figure 1 for a simplified description of the process.

The main result of this section is as follows.

**Theorem 1.** *The following are true for Algorithms 2 and 3,*

---

**Algorithm 3** TreeUnlearn

---

**Input:** $z_j$: data point to be deleted, , $\mathcal{T}$: internal tree data-structure saved during learning
 1: $s = \mathsf{leaf}(j)$ and $l = \mathsf{leaf}(|S|)$
 2: $(\cdot, \cdot, w, z) = \mathsf{Get}(s; \mathcal{T})$ and $(\cdot, \cdot, \cdot, z') = \mathsf{Get}(l; \mathcal{T})$
 3: $g = p_j(\{w_q\}_{q \leq s}, z)$ and $g' = p_j(\{w_q\}_{q \leq s}, z')$
 4: Let path $= \{l \to \cdots \to \mathrm{root}\}$ be the path from $l$ to root.
 5: **for** $b \in$ path **do** $u_b = u_b - g'$ **end for**
 6: Remove node $l$ from $\mathcal{T}$.
 7: Let $b = s$ and ct $= 1$
 8: **if** $j = |S|$ **then** let $b = \emptyset$ **end if**
 9: **while** $b \neq \emptyset$ **do**
10: $\quad (u, r, \cdot, \cdot) = \mathsf{Get}(b; \mathcal{T})$
11: $\quad u' = u - g + g'$
12: $\quad$ **if** $\mathsf{Unif}\,(0,1) \leq \frac{\phi_{\mathcal{N}(u, \sigma^2 \mathbb{I})}(r)}{\phi_{\mathcal{N}(u', \sigma^2 \mathbb{I})}(r)}$ **then**
13: $\qquad$ **if** $b = s$ **then** $\mathsf{Set}(b, (u', r, w, z'); \mathcal{T})$
$\qquad\quad$ **else then** $\mathsf{Set}(b, (u', r, \emptyset, \emptyset); \mathcal{T})$ **end if**
14: $\quad$ **else**
15: $\qquad r' = \mathsf{Reflect}(u, u', r)$
16: $\qquad$ **if** $b = s$ **then**
17: $\qquad\quad \mathsf{Set}(b; (u', r', \emptyset, z'); \mathcal{T})$
18: $\qquad\quad w' = U_j\left(\{w_q\}_{q \leq b}, \mathsf{GetPrefixSum}(j; \mathcal{T})\right)$
19: $\qquad\quad \mathsf{Set}(b, (u', r', w', z'); \mathcal{T})$
20: $\qquad$ **else**
21: $\qquad\quad \mathsf{Set}(b, (u', r', \emptyset, \emptyset); \mathcal{T})$
22: $\qquad$ **end if**
23: $\qquad \mathsf{TreeLearn}(j + \mathrm{ct}; \mathcal{T})$ // *Continue Retraining*
24: $\qquad$ **break**
25: $\quad$ **end if**
26: $\quad$ **if** $b$ is left sibling **then** ct $= \mathrm{ct} + 2^{|s| - |b| - 1}$ **end if**
27: $\quad$ Set (new) $b$ as binary representation of parent of $b$
28: **end while**
29: Update dataset $S = S \backslash \{z_j\}$
**Output:** $\widehat{w} = \mathcal{S}(\{w_b\}_b)$

---

1. *The learning Algorithm 2 with $\sigma^2 = \frac{64B^2 log^2(n)}{\rho}$ satisfies $\rho$-TV stability.*

2. *The corresponding unlearning Algorithm 3 satisfies exact unlearning.*

3. *The relative unlearning complexity is $\widetilde{O}(\rho)$*

As discussed in the preceding section, in the Theorem above, we have all the properties we needed with the unlearning process. We now move on to applications and give accuracy guarantees.

## 5 APPLICATIONS

In the following, we describe some problems and learning algorithms. The corresponding unlearning algorithms and its correctness simply follow as application of the result of the preceding section, provided we show that it uses a bounded sensitivity prefix-sum query. The only other thing to show is the accuracy guarantee of the TV stable modification of the learning algorithm (Algorithm 2).

We use runtime to mean gradient complexity as is standard in convex optimization (Nemirovskij & Yudin, 1983). But, as pointed out before, our proposed unlearning algorithm yields similar improvements over retraining, even accounting for other operations in the method.

### 5.1 SMOOTH SCO WITH VARIANCE REDUCED FRANK-WOLFE

We assume that the loss function $w \mapsto \ell(w; z)$ is $H$-smooth and $G$-Lipschitz for all $z$[1]. The algorithm we use is variance reduced Frank-Wolfe method where the variance reduced gradient estimate $u_t$ is the Hybrid-SARAH estimate (Tran-Dinh et al., 2019) with $\gamma_t = \frac{1}{t+1}$ given as,

---

[1] A real valued function $w \mapsto f(x)$ is $G$-Lipschitz and $H$-smooth if $|f(x_1) - f(x_1)| \leq G \|x_1 - x_2\|$ an $\|\nabla f(x_1) - \nabla f(x_2)\| \leq H \|x_1 - x_2\|$ respectively.

$$u_t = (1 - \gamma_t)\left(u_{t-1} + \nabla\ell(w_t; z_t) - \nabla\ell(w_{t-1}; z_t)\right) + \gamma_t \nabla\ell(w_t; z_t)$$

$$= \frac{1}{t+1}\sum_{i=1}^{t}\left((i+1)\nabla\ell(w_i; z_i) - i\nabla\ell(w_{i-1}; z_i)\right)$$

We show that the above is a prefix sum query with sensitivity $B = 2(HD + G)$, thus fits into our framework. The full pseudo-code is given as Algorithm 12 in Appendix E. We state the main result below where the accuracy guarantee follows from modifications to the analysis in Zhang et al. (2020).

**Theorem 2.** *Let $\rho \leq 1$ and $\ell : \mathcal{W} \times \mathcal{Z} \rightarrow \mathbb{R}$ be an $H$-smooth, $G$-Lipschitz convex function over a convex set $\mathcal{W}$ of diameter $D$. Algorithm 12, as the learning algorithm, run with $\sigma^2 = \frac{64(HD+G)^2 log^2(n)}{\rho^2}$, $t_0 = 1$ and $\eta_t = \frac{1}{t+1}$ on a dataset $S$ of $n$ i.i.d. samples from $\mathcal{D}$ outputs $\widehat{w}$, with excess population risk bounded as, $\mathbb{E}\left[L(\widehat{w}; \mathcal{D}) - L(w^*; \mathcal{D})\right] = \widetilde{O}\left((G + HD)D\left(\frac{1}{\sqrt{n}} + \frac{\sqrt{d}}{n\rho}\right)\right)$. Furthermore, the corresponding unlearning Algorithm 3 (with query and update functions as specified in the learning algorithm), satisfies exact unlearning with $\widetilde{O}(\rho n)$ expected runtime.*

### 5.2 NON-SMOOTH SCO WITH DUAL AVERAGING

In this section, we only assume that loss function $w \mapsto \ell(w; z)$ is $G$-Lipschitz and convex $\forall z \in \mathcal{Z}$. Herein, we use dual averaging method (Nesterov, 2009) where the model is updated as follows: $w_{t+1} = \Pi_{\mathcal{W}}\left(w_0 - \eta\sum_{i=1}^{t}\nabla\ell(w_i; z_i)\right)$, where $\Pi$ denotes the Euclidean projection on to the convex set $\mathcal{W}$. The above again is a prefix-sum query with sensitivity $G$, thus fits into our framework. The full pseudo-code is given as Algorithm 13 in Appendix E. The accuracy guarantee mainly follows from Kairouz et al. (2021).

**Theorem 3.** *Let $\rho \leq 1$ and $\ell : \mathcal{W} \times \mathcal{Z} \rightarrow \mathbb{R}$ be a $G$-Lipschitz convex function over a convex set $\mathcal{W}$ of diameter $D$. Algorithm 13, as the learning algorithm, run with $\sigma^2 = \frac{64G^2 log^2(n)}{\rho^2}$, $t_0 = 1$ and $\eta = \frac{Dd^{1/4}\sqrt{\log(n)}}{G\sqrt{n\rho}}$ on a dataset $S$ of $n$ samples, drawn i.i.d. from $\mathcal{D}$, outputs $\widehat{w}$ with excess population risk bounded as $\mathbb{E}\left[L(\widehat{w}; \mathcal{D}) - L(w^*; \mathcal{D})\right] = \widetilde{O}\left(GD\left(\frac{1}{\sqrt{n}} + \sqrt{\frac{\sqrt{d}}{n\rho}}\right)\right)$. Furthermore, the corresponding unlearning Algorithm 3 (with query and update functions as specified in the learning algorithm), satisfies exact unlearning with $\widetilde{O}(\rho n)$ expected runtime.*

### 5.3 CONVEX GLM WITH JL METHOD

---

**Algorithm 4** JL method

---

**Input:** Dataset $S$, loss function $\ell$, Algorithm $\mathcal{A}$, JL matrix $\Phi \in \mathbb{R}^{d \times k}$, Noise variance $\sigma^2$
1: $\Phi S = \{\Phi x_i\}_{i=1}^{n}$
2: $\widetilde{w} = \mathcal{A}(\ell, \Phi S, 2G\|\mathcal{X}\|, 2H\|\mathcal{X}\|^2, \sigma)$
**Output:** $\widehat{w} = \Phi^\top \widetilde{w}$

---

This JL method, proposed in Arora et al. (2022), is a general technique to get dimension-independent rate for unconstrained convex GLMs from algorithms giving dimension-dependent rate for constrained (general) convex losses. The method, described in Algorithm 4, simply embeds the dataset into a low dimensional space, via a JL matrix $\Phi$, and then runs a base algorithm on the low dimensional dataset.

**Smooth, Lipschitz GLMs:** We assume that $\phi_y : \mathbb{R} \rightarrow \mathbb{R}$ is convex, $H$-smooth and $G$-Lipschitz for all $y \in \mathcal{Y}$. Using VR-Frank Wolfe as the base algorithm, we get a rate of $O\left(\frac{1}{\sqrt{n}} + \frac{1}{(n\rho)^{2/3}}\right)$ with relative unlearning efficiency of $\rho$ – see Theorem 5 for a precise statement.

**Lipschitz GLMs:** We assume that $\phi_y : \mathbb{R} \rightarrow \mathbb{R}$ is convex and $G$-Lipschitz for all $y \in \mathcal{Y}$. We give the following result in this case using Dual Averaging as the base algorithm. Using Dual Averaging as the base algorithm, we get a rate of $O\left(\frac{1}{\sqrt{n}} + \frac{1}{(n\rho)^{1/3}}\right)$ with relative unlearning efficiency of $\rho$ – see Theorem 6 for the precise statement.

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

## A    ADDITIONAL PRELIMINARIES

We recall some concepts from differential privacy which will be useful in our algorithmic techniques.

**Definition 7.** *An algorithm $\mathcal{A}$ satisfies $(\alpha, \epsilon(\alpha))$-Rényi Differential Privacy (RDP), if for any two datasets $S$ and $S'$ which differ in one data point ($|S \Delta S'| = 1$), the $\alpha$-Rényi Divergence between $\mathcal{A}(S)$ and $\mathcal{A}(S')$, with probability densities $\phi_{\mathcal{A}(S)}$ and $\phi_{\mathcal{A}(S')}$, defined as follows:*

$$D_\alpha\left(\mathcal{A}(S)\|\mathcal{A}(S')\right) = \frac{1}{\alpha - 1} \ln\left(\int_{Range(\mathcal{A})} \phi_{\mathcal{A}(S)}(x)^\alpha \phi_{\mathcal{A}(S')}(x)^{1-\alpha} dx\right)$$

*is bounded as, $D_\alpha(\mathcal{A}(S)\|\mathcal{A}(S')) \leq \epsilon(\alpha)$.*

RDP satisfies many desirable properties such as adaptive and parallel composition and amplification by sub-sampling (Mironov, 2017; Wang et al., 2019). Furthermore, we give the following lemma which relates TV stability to RDP.

**Lemma 1** (RDP $\implies$ TV-stability)**.** *If an algorithm satisfies $(\alpha, \epsilon(\alpha))$-RDP, then it satisfies $\left(1 - \exp\left(-\lim_{\alpha\downarrow 1} \epsilon(\alpha)\right)\right)^{\frac{1}{2}}$-TV stability.*

*Proof of Lemma 1.* From Theorem 4 in Van Erven & Harremos (2014), we have that $\lim_{\alpha\downarrow 1} D_\alpha(P\|Q) = $ KL $(P\|Q)$, where $KL(\cdot\|\cdot)$ denotes the Kullback-Leibler (KL) divergence between the two distributions. Finally, we relate the TV distance with the KL divergence using Bretagnolle–Huber bound (Bretagnolle & Huber, 1979; Canonne, 2022) which gives the claimed bound. $\qquad\square$

## B    UNLEARNING FOR LINEAR QUERIES

A basic form of a query we consider is a linear query, defined as follows.

**Definition 8.** *A query $q : \mathcal{W}^* \times \mathcal{Z}^n \to \mathcal{W}$ is a linear query if $q(\{w_i\}_i; S) = \sum_{j\in S} p_j(\{w_i\}_i; z_j)$ for some functions $p_j : \mathcal{W}^* \times \mathcal{Z} \to \mathcal{W}$.*

We consider the class of $B$-sensitive linear queries. We give the TV stable modified learning procedure in Algorithm 5 which basically releases the linear queries perturbed with Gaussian noise of appropriate variance.

---

**Algorithm 5** LearnLinearQueries($w_{t_0}, t_0$)

---

**Input:** Dataset $S$, initial iteration $t_0$, steps $T$, query functions $\{q_t(\cdot)\}_{t\leq T}$, update functions $\{U_t(\cdot)\}_{t\leq T}$, Selector function $\mathcal{S}(\cdot)$, noise variance $\sigma$
1: Initialize model $w_1 \in \mathcal{W}$
2: **for** $t = t_0$ to $T - 1$ **do**
3:     Query the dataset $u_t = q_t\left(\{w_i\}_{i\leq t}; S\right)$.
4:     Perturb: $r_t = u_t + \xi_t$ where $\xi_t \sim \mathcal{N}(0, \sigma^2 \mathbb{I}_d)$.
5:     Update $w_{t+1} = U_t(\{w_i\}_{i\leq t}, r_t)$
6:     Save $(u_t, r_t, w_{t+1})$
7: **end for**
**Output:** $\widehat{w} = \mathcal{S}\left(\{w_t\}_{t\leq T}\right)$

---

Note that the underlying probability distribution that the above learning algorithm samples from is a Markov chain. The corresponding unlearning procedure, described in Algorithm 6, is based on constructing a coupling between the Markov chains for the current dataset and the dataset without the to-be-deleted point. In particular, we start from the first iteration, perform rejection sampling, if it results in acceptance, then we proceed to the second iteration and so on. If some iteration results in rejection, then we do the reflection step, and continue retraining from there on.

---

**Algorithm 6** Unlearning algorithm for linear queries

---

**Input:** Deleted point $z_j$,
1: **for** $t = 1$ to $T - 1$ **do**
2:    $(u_t, r_t, w_t) = \mathsf{Load}\,()$
3:    Compute $u'_t = u_t - p_t^j\left(\{w_i\}_{i \leq t}\,;z_j\right)$
4:    **if** $\mathsf{Unif}\,(0,1) \leq \frac{\phi_{\mathcal{N}(u_t, \sigma^2 \mathbb{I})}(r_t)}{\phi_{\mathcal{N}(u'_t, \sigma^2 \mathbb{I})}(r_t)}$ **then**
5:      $\mathsf{Save}\,(u'_t)$
6:    **else**
7:      $r'_t = \mathsf{reflect}(r_t, u_t, u'_t)$
8:      $w_{t+1} = U_t\left(\{w_i\}_{i \leq t}\,, r'_t\right)$
9:      $\mathsf{LearnLinearQueries}(w_{t+1}, t+1)$
10:      **break**
11:    **end if**
12: **end for**

---

The above is basically the same unlearning algorithm as that of Ullah et al. (2021) but presented in the general context of linear queries. Therefore, it generalizes the framework of Ullah et al. (2021) which was limited to the Stochastic Gradient Descent algorithm. We also remark that linear queries can often be augmented with a sub-sampling operator yielding *amplified* guarantees, as done in Ullah et al. (2021). However, we omit this extension for brevity. The main result of this section is as follows.

**Theorem 4.** *The following are true for Algorithms 5 and 6,*

1. *The learning algorithm, Algorithm 5 with $\sigma^2 = \frac{64 B^2}{n^2 \rho^2}$ satisfies $\rho$-TV stability.*

2. *The unlearning algorithm, Algorithm 6, corresponding to Algorithm 5, satisfies exact unlearning.*

3. *The relative unlearning complexity is $O\left(\rho\sqrt{T}\right)$.*

*Proof.* This proof simply follows from the observation that the analysis of Ullah et al. (2021) only uses the bounded sensitivity linear query structure of the stochastic gradient method for their TV stability bound as well as correctness and runtime of the unlearning procedure. □

### B.1 Applications

This generalization yields the following applications.

### B.2 Federated unlearning for Federated Averaging

In the federated learning setting, we have $C$ clients (which typically correspond to user devices) with their own datasets and a parameter server (aggregator). A typical, informal, goal is training a single globally shared model using all the dataset with small communication between the clients and the server, and without moving any private data (explicitly) to the server. Federated Averaging (Konecnỳ et al., 2016), described in Algorithm 7, is a widely used method in federated learning. Note that in the every round of the method, the client outputs, $\{w_t^c\}_{c=1}^C$, are aggregated using an averaging operation:

$$w_t = \frac{1}{C} \sum_{c=1}^{C} w_t^c.$$

In Algorithm 7, $\mathsf{ClientUpdate}$ is a function which runs on the client's data using the current model $w_t$ and problem specific-parameter $\mathcal{P}$ (such as as number of steps, learning rate of some optimization routine). For brevity, we do not instantiate the $\mathsf{ClientUpdate}$ function, but usually some variant of stochastic gradient descent is used.

---

**Algorithm 7** Federated Averaging (Server side)

---

**Input:** Number of clients $C$, number of rounds $T$, client-specific parameters $\mathcal{P}$
1: Initialize model $w_1 \in \mathcal{W}$
2: **for** $t = 1$ to $T - 1$ **do**
3:      **for** $c = 1$ to $C$ **do**
4:          $w_{t+1}^c = \mathsf{ClientUpdate}\,(c, w_{t-1}, \mathcal{P})$
5:      **end for**
6:      $w_{t+1} = \frac{1}{C} \sum_{c=1}^C w_{t+1}^c$
7: **end for**
**Output:** $\widehat{w} = \mathcal{S}\left( \{w_t\}_{t \leq T} \right)$

---

**Federated Unlearning:** In the federated unlearning problem, after a model is trained, one of the clients requests to remove themselves from the process. The parameter server then needs to update the model (and state) in such a way that it is indistinguishable to the state if the client were absent. Hence, this is analogous to the standard unlearning problem with the client playing the role of a data point. This analogy also occurs with private federated learning wherein the *widely-used* granularity of differential privacy is user-level differential privacy (McMahan et al., 2017). In this case, a client (potentially containing multiple data items) plays the role of a data item, the presence/absence of which is used in the differential privacy definition.

**TV-stable learning and unlearning:** The model aggregation step (line 6 in Algorithm 7) of the federated averaging method is a linear query over the clients. Moreover, if the clients output models that are bounded in norm, then it is a bounded sensitivity linear query (typically enforced by clipping the updates). Hence, this fits into the template of linear query release method and thus can be modified, as in Algorithm 5 to be TV stable. The corresponding unlearning method is the one given in Algorithm 6.

### B.3    Lloyd's algorithm for $k$-means clustering

In this section, we briefly discuss how an algorithm for $k$-means clustering fits into the linear query release framework. We remark that the prior work of Ginart et al. (2019) gave an unlearning method for this problem based on randomized quantization, which can also be seen as a specific TV-stable algorithm followed by a coupling based unlearning method.

Lloyd's algorithm is a widely used method for $k$-means clustering. Herein, starting with an arbitrary choice of centers, we construct a partition of the dataset, which thereby gives a new set of centers. This process is repeated for a certain number of rounds. The method is described as Algorithm 8.

We notice again that the updates for every cluster, line 7 in Algorithm 8, is a linear query, hence it fits into the linear query release template and thus learning and unlearning algorithms based on linear queries readily follow.

---

**Algorithm 8** Lloyd's algorithm

---

**Input:** Number of clusters $C$, number of rounds $T$, dataset $S = \{z_i\}_{i=1}^n$.
1: Initialize centers $\{w_c\}_{c=1}^C$
2: **for** $t = 1$ to $T - 1$ **do**
3:      **for** $c = 1$ to $C$ **do**
4:          Compute $S_c = \left\{ z_1^c, z_2^c, \ldots z_{|S_c|}^c \right\}$, the set of data-points closest to $w_c$.
5:      **end for**
6:      **for** $c = 1$ to $C$ **do**
7:          Update $w_c = \frac{1}{|S_c|} \sum_{i=1}^{|S_c|} z_i^c$
8:      **end for**
9: **end for**
**Output:** $\{w_c\}_{c=1}^C$

---

## C   Missing Details from Section 4

In this section, we provide pseudo-code of the operations supported by the binary tree data structure.

---

**Algorithm 9** Append$(u, \sigma; \mathcal{T})$

---

**Input:** Query response $u$, noise variance $\sigma$,Tree $\mathcal{T}$
1: Let $s$ be the first empty leaf index
2: path $= \{s \rightarrow \cdots \text{root}\}$ be the path from $s$ to root.
3: UpdateTree$(u, \text{path}, \sigma; \mathcal{T})$

---

**Algorithm 10** UpdateTree$(u, \text{path}, \sigma; \mathcal{T})$

---

**Input:** Query response $u$, Set of nodes path, noise variance $\sigma$,Tree $\mathcal{T}$
1: **for** $b \in$ path **do**
2:     $u_b = u_b + u$
3:     **if** $b$ is a left child or $b$ is a leaf **then**
4:         $\xi \sim \mathcal{N}(0, \sigma^2 \mathbb{I})$
5:         $r_b = u_b + \xi$
6:         **break**
7:     **end if**
8: **end for**

---

**Algorithm 11** GetPrefixSum$(t; \mathcal{T})$

---

**Input:** $t \in \mathbb{N}$, Tree $\mathcal{T}$,
1: Initialize $g \in \mathbb{R}^p$ to 0
2: $s \leftarrow \text{leaf}(t)$
3: Let path be the path from $s$ to root.
4: **while** $b \neq \emptyset$ **do**
5:     **if** $b$ is a leaf child or $b$ is a leaf **then**
6:         $g = g + r_b$
7:     **end if**
8: **end while**
**Output:** $g$

---

## D   Missing Proofs from Section 4

*Proof of Theorem 1.* The first part of the Theorem follows from Lemma 2 followed by post-processing to argue that the same TV stability parameter holds for the final iterate.

The second part, exact unlearning, follows from Lemma 5 wherein $Q$ denotes the distribution of the algorithm's output run on the dataset without the to-be-deleted point.

For the third part, note that the unlearning algorithm 3 makes two queries if no retraining is triggered. If a retraining is triggered, the number of queries it makes is at most the query complexity of learning algorithm, $T = n$. Finally, the probability of retraining, from Lemma 6 is at most $\log(n)\rho$. Combining, this gives the stated bound on relative unlearning complexity.                    □

### D.1   Lemmas for Unlearning

**Additional notation:** We first present some additional notation used in the statement and proof of the following lemmas. Let $S$ and $S'$ be datasets before and after the unlearning request. Let $P$ and $Q$ denote the probability measures over the range of tree data-structure, which is $\mathfrak{T} = \left(\mathbb{R}^d \times \mathbb{R}^d \times \mathbb{R}^d \times [n]\right)^n$, induced by the output of learning algorithm on $S$ and $S'$ respectively. We order the nodes of the binary tree w.r.t. the post-order traversal of tree. Hence, given two nodes $v$ and $v'$ or their binary representations $s$ and $s'$, we use $v \leq v$ or $s \leq s'$ w.r.t the above ordering. Given a

node $b$, let $P_b\left(\cdot|\mathcal{T}_{\leq b}\right)$ denote the conditional distribution of the nodes given the prefix nodes of the tree.

Let $\mathbf{p}$ be a permutation over $[n]$ and $p_b$ denote the index on the $b$-th node, when $b$ is a leaf. Let $\mu$ denote the probability, and conditional probability, depending on context, of $\mathbf{p}$ and $p_b$, under the random permutation model. Specifically, we use $\mu(\mathbf{p})$ and $\mu(p_b|p_{\leq b})$ to denote the probability of the sequence $\mathbf{p}$ and conditional probability of $p_b$ given the previous values.

Let $\mathcal{T}^{(1)}$ denote the initial binary tree i.e. the one constructed after the algorithm is run on dataset $S$, and $\mathcal{T}^{(2)}$ be the binary tree constructed after unlearning. Let $P^{\mathbf{p}}$ and $Q^{\mathbf{p}}$ denote the conditional distributions for $P$ and $Q$ respectively given permutation $\mathbf{p}$.

We factor the probability density of $P$ as:
$$\phi_P\left(\mathcal{T}^{(1)}\right) = \prod_{b \in B} \phi_{P_b}\left(v_b^{(1)}|\mathcal{T}_{\leq b}^{(1)}\right) = \prod_{b \in B} \mu(p_b^{(1)}|p_{\leq b}^{(1)})\phi_{P_b^{p_{\leq b}^{(1)}}}\left(u_b^{(1)}, r_b^{(1)}, w_b^{(1)}|\mathcal{T}_{\leq b}^{(1)}\right)$$

Fixing the permutation sequence $\mathbf{p}^{(1)}$, denote and factor the conditional distribution as,
$$\phi_P^{\mathbf{p}^{(1)}}(\mathcal{T}^{(1)}) = \prod_{b \in B} \phi_{P_b^{\mathbf{p}_{\leq b}^{(1)}}}\left(u_b^{(1)}, r_b^{(1)}, w^{(1)}|\mathcal{T}_{\leq b}^{(1)}\right)$$

Finally, define response trees $\widetilde{\mathcal{T}}^{(1)}$ and $\widetilde{\mathcal{T}}^{(2)}$ which only contain the response variables $(r_b)_b$. Moreover, define distributions $\widetilde{P}, \widetilde{P}_b, \widetilde{P}^{\mathbf{p}}, \widetilde{P}_b^{\mathbf{p}}$ and $\widetilde{Q}, \widetilde{Q}_b, \widetilde{Q}^{\mathbf{p}} \widetilde{Q}_b^{\mathbf{p}}$ as before.

We first show the the tree $\widetilde{\mathcal{T}}$ produced by the learning algorithm is TV-stable.

**Lemma 2.** *Let $0 < \rho \leq 1, B \geq 0, n \in \mathbb{N}$. For $B$-sensitive prefix sum queries, setting $\sigma^2 = \frac{64B^2 log^2(n)}{\rho^2}$, the response tree data structure $\widetilde{\mathcal{T}}$ is $\rho$-TV stable.*

*Proof.* The proof of privacy of tree aggregation is classical in differential privacy, see Guha Thakurta & Smith (2013) for example. The proof has three ingredients: Gaussian mechanism guarantee, parallel composition (to argue that accounting along the height of the tree suffices) and adaptive composition (for accounting along the height of the tree). Since the noise is Gaussian and these composition properties also holds under RDP (Mironov, 2017), therefore we can give an RDP guarantee of $\epsilon(\alpha) \leq \log^2(n) \cdot \frac{64\alpha B^2}{\sigma^2}\alpha\rho^2$. Finally, using Lemma 1 and a numerical simplification since $\rho \leq 1$ gives the claimed result. $\square$

Recall that $j$ is the index of the data item (after permutation) which is deleted. Without loss of generality, assume that the original index of the deleted data-point is $n$. We first argue the following about the distribution of $\mathbf{p}^{(1)}$ and $\mathbf{p}^{(2)}$.

**Lemma 3.** *For any set $E \subseteq [n]^n$ and any set $E' \subseteq [n-1]^{n-1}$, we have*
$$\mathbb{P}_{\mathbf{p}^{(1)}}\left(\mathbf{p}^{(1)} \in E\right) = \mu_n(E)$$
$$\mathbb{P}_{\mathbf{p}^{(2)}}\left(\mathbf{p}^{(2)} \in E'\right) = \mu_{n-1}(E')$$

*Proof.* Since $\mathbf{p}^{(1)}$ and $\mathbf{p}^{(2)}$ are discrete distributions, it suffices to argue the above for the atoms. Firstly, by construction, $\mathbf{p}^{(1)} \sim \mu_n$ and hence the first part is done. For the second part for any sequence $h = (h_i)_{i=1}^{n-1}$ where $h_i \in [n-1]$. Let $[h, j]$ denote the concatenation of $h$ and $j$ (the deleted index). By symmetry, the probability
$$\mathbb{P}_{\mathbf{p}^{(2)}}(h) = \frac{1}{n+1}\mathbb{P}_{\mathbf{p}^{(1)}}([h, j]) = \mu_{n-1}(h)$$
This completes the proof. $\square$

We now show transport of the conditional distribution by the unlearning operation.

**Lemma 4.** *For any measurable event $E \subseteq \mathbb{R}^{d|\mathcal{T}^{(2)}|}$,*
$$\mathbb{P}\left(\widetilde{\mathcal{T}}^{(2)} \in E|\mathbf{p}^{(1)}, \mathbf{p}^{(2)}\right) = \widetilde{Q}^{\mathbf{p}^{(2)}}(E).$$

*Proof.* The proof is based on induction on the nodes of $\widetilde{\mathcal{T}}^{(2)}$ in the post-order traversal. Let $\left(v_b^{(1)}\right)_b$ and $\left(v_b^{(2)}\right)_b$ be the nodes of the tree arranged in the post-order traversal order. Given $j$, index of the item deleted, let $s = \mathsf{leaf}(j)$. Define $\mathsf{prefix}(s)$ and $\mathsf{suffix}(s)$, as set of nodes before and after $s$ respectively in the $\leq$ order.

Given an event $E \subseteq \mathbb{R}^{d|\widetilde{\mathcal{T}}^{(2)}|}$ and $r_{\leq b}$, define $E_b^{r_{\leq b}}$ as follows:

$$E_b^{r_{\leq b}} = \left\{ e \in \mathbb{R}^d : \exists \overline{e} \in \left( \times_{>b} \mathbb{R}^d \right) : (r_{\leq b}, e, \overline{e}) \in E \right\}$$

where $\times_{>b} \mathbb{R}^d$ denote the Cartesian product of $\mathbb{R}^d$'s of upto $> b$ but smaller than or equal to $\left| \mathcal{T}^{(1)} \right|$ elements. Similarly, define $E_{\geq b}^{r_{\leq b}}$ as,

$$E_{\geq b}^{r_{\leq b}} = \left\{ \mathbf{e} \in \left( \times_{\geq b} \mathbb{R}^d \right) : (r_{\leq b}, \mathbf{e}) \in E \right\}$$

Finally, define $E_{<b}$ as

$$E_{<b} = \left\{ \mathbf{e} \in \left( \times_{<b} \mathbb{R}^d \right) : \exists \overline{\mathbf{e}} \in \left( \times_{\geq b} \mathbb{R}^d \right) : (\mathbf{e}, \overline{\mathbf{e}}) \in E \right\}$$

We now factorize the probability below as,

$$
\begin{aligned}
\mathbb{P}\left( \widetilde{\mathcal{T}}^{(2)} \in E | \mathbf{p}^{(1)}, \mathbf{p}^{(2)} \right) &= \prod_{b \in \mathsf{prefix}(s)} \mathbb{P}\left( r_b^{(2)} \in E_b^{r_{<b}^{(2)}} | p_b^{(2)}, r_{<b}^{(2)} \right) \mathbb{P}\left( \widetilde{\mathcal{T}}_{\geq s}^{(2)} \in E_{\geq s}^{r_{<s}^{(2)}} | \widetilde{\mathcal{T}}_{<s}^{(2)}, \mathbf{p}^{(1)}, \mathbf{p}^{(2)} \right) \\
&= \prod_{b \in \mathsf{prefix}(s)} \mathbb{P}\left( r_b^{(1)} \in E_b^{r_{<b}^{(1)}} | p_b^{(1)}, r_{<b}^{(1)} \right) \mathbb{P}\left( \widetilde{\mathcal{T}}_{\geq s}^{(2)} \in E_{\geq s}^{r_{<s}^{(2)}} | \widetilde{\mathcal{T}}_{<s}^{(2)}, \mathbf{p}^{(1)}, \mathbf{p}^{(2)} \right) \\
&= \prod_{b \in \mathsf{prefix}(s)} P_b \left( E_b^{r_{<b}^{(1)}} | p_b^{(1)}, r_{<b}^{(1)} \right) \mathbb{P}\left( \widetilde{\mathcal{T}}_{\geq s}^{(2)} \in E_{\geq s}^{r_{<s}^{(2)}} | \widetilde{\mathcal{T}}_{<s}^{(2)}, \mathbf{p}^{(1)}, \mathbf{p}^{(2)} \right) \\
&= \prod_{b \in \mathsf{prefix}(s)} Q_b \left( E_b^{r_{<b}^{(2)}} | p_b^{(2)}, r_{<b}^{(2)} \right) \mathbb{P}\left( \widetilde{\mathcal{T}}_{\geq s}^{(2)} \in E_{\geq s}^{r_{<s}^{(2)}} | \widetilde{\mathcal{T}}_{<s}^{(2)}, \mathbf{p}^{(1)}, \mathbf{p}^{(2)} \right) \\
&= Q_{<s} \left( E_{<s} | p_{<s}^{(2)}, r_{<s}^{(2)} \right) \mathbb{P}\left( \widetilde{\mathcal{T}}_{\geq s}^{(2)} \in E_{\geq s}^{r_{<s}^{(2)}} | \widetilde{\mathcal{T}}_{<s}^{(2)}, \mathbf{p}^{(1)}, \mathbf{p}^{(2)} \right)
\end{aligned}
$$

where the second equality follows since $r_{\leq b}^{(1)} = r_{\leq b}^{(2)}$ and $p_b^{(1)} = p_b^{(2)}$ for all $b < s$ by construction. The third equality follows since $r_b^{(1)}$ is distributed as $P_b$ conditionally and fourth and final follows since conditioned on the permutation being the same, the prefix is also distributed as $Q_{<s}$.

We now start the induction: let $I$(induction variable) be $I = s$ i.e the last item is deleted. In this case, the unlearning algorithm simply removes the $s$-th node of the tree and all we are left with is the tree with $\mathsf{prefix}(s)$ nodes, which as argued above is distributed as $Q_{<s} = Q$.

For the case $I = s + 1$: we simply focus on $\widetilde{\mathcal{T}}_{\geq s}^{(2)} = \widetilde{\mathcal{T}}_s^{(2)} = r_s^{(2)}$. Note that $r_s^{(1)}$ is distributed as $\mathcal{N}(u^{(1)}, \sigma^2 \mathbb{I})$ and we want $r_s^{(2)}$ distributed as $\mathcal{N}(u^{(2)}, \sigma^2 \mathbb{I})$. The operation in the algorithm is basically a one step reflection coupling which from Lemma 1 in Ullah et al. (2021) satisfies,

$$\mathbb{P}\left( r_s^{(2)} \in E_s^{r_{<s}^{(2)}} | \mathbf{p}^{(1)}, \mathbf{p}^{(2)} \right) = Q_s^{p_s^{(2)}} \left( E_s^{r_{<s}^{(2)}} \right)$$

Therefore,

$$\mathbb{P}\left( \widetilde{\mathcal{T}}^{(2)} \in E | \mathbf{p}^{(1)}, \mathbf{p}^{(2)} \right) = Q_{<s} \left( E_{<s} | p_{<s}^{(2)}, r_{<s}^{(2)} \right) \widetilde{Q}_s^{p_s^{(2)}} \left( E_s^{r_{<s}^{(2)}} \right) = \widetilde{Q}^{\mathbf{p}^{(2)}}(E)$$

This finishes the base cases.

We now proceed to the induction step: suppose the following claim holds for nodes upto $I = k$ – for any event $E$, the marginal distribution

$$\mathbb{P}\left( \mathcal{T}_{\leq k}^{(2)} \in E | \mathbf{p}^{(1)}, \mathbf{p}^{(2)} \right) = \widetilde{Q}_{\leq k} \left( E | \mathbf{p}^{(2)} \right)$$

For node $k + 1$, consider a few cases:

1. A: All rejection sampling steps prior to node $k$ resulted in accepts:

    (a) AP: Node $k+1$ lies in the path from the $s$ to root.

        i. APA: The rejection sampling at this node succeeds.

        ii. APR: The rejection sampling at this node fails i.e. a reflection step is performed.

    (b) AN: Node $k+1$ doesn't lie in the path from $s$ root.

2. R: Some rejection sampling step resulted in rejection.

For case R, we have that $r_{k+1}^{(2)} \sim \widetilde{Q}_{k+1}(\cdot|\widetilde{\mathcal{T}}_{\leq k}^{(2)}, \mathbf{p}^{(2)})$. For the case AN, note that the random variable $r_{k+1}^{(2)} = r_{k+1}^{(1)}$, hence,

$$\mathbb{P}\left(r_{k+1}^{(2)} \in E_{k+1}^{r_{\leq k}^{(2)}}|\mathsf{AN}, \mathcal{T}_{\leq k}^{(2)}, \mathbf{p}^{(1)}, \mathbf{p}^{(2)}\right) = \widetilde{P}_{k+1}\left(E_{k+1}^{r_{\leq k}^{(2)}}|\mathbf{p}^{(2)}, \widetilde{\mathcal{T}}_{\leq k}^{(2)}\right) = \widetilde{Q}_{k+1}\left(E_{k+1}^{r_{\leq k}^{(2)}}|\mathbf{p}^{(2)}, \widetilde{\mathcal{T}}_{\leq k}^{(2)}\right)$$

where the last equality follows since the dependence of $r_{k+1}^{(2)}$ is only on data points which are leaves of the sub-tree rooted at node $k+1$. These, by assumption do not contain the data point $s$, hence is identically distributed as $P_{k+1}$.

For the event AP, we have,

$$\mathbb{P}\left(r_{k+1}^{(2)} \in E_{k+1}^{r_{\leq k}^{(2)}}|\mathsf{AP}, \mathbf{p}^{(1)}\mathbf{p}^{(2)}, \widetilde{\mathcal{T}}^{(2)}\right) = \mathbb{P}\left(r_{k+1}^{(2)} \in E_{k+1}^{r_{\leq k}^{(2)}}, \mathsf{APA}|\mathsf{AP}, \mathbf{p}^{(1)}, \mathbf{p}^{(2)}, \widetilde{\mathcal{T}}_{\leq k}^{(2)}\right)$$

$$+ \mathbb{P}\left(r_{k+1}^{(2)} \in E_{k+1}^{r_{\leq k}^{(2)}}, \mathsf{APR}|\mathsf{AP}, \mathbf{p}^{(1)}, \mathbf{p}^{(2)}, \widetilde{\mathcal{T}}_{\leq k}^{(2)}\right)$$

$$= \widetilde{Q}_{k+1}\left(E_{k+1}^{r_{\leq k}^{(2)}}|\mathbf{p}^{(1)}, \mathbf{p}^{(2)}, \widetilde{\mathcal{T}}_{\leq k}^{(2)}\right)$$

where the last step follows from Lemma 1 in Ullah et al. (2021) .

Hence, combining AP and AN cases,

$$\mathbb{P}\left(r_{k+1}^{(2)} \in E_{k+1}^{r_{\leq k}^{(2)}}|\mathsf{AN}, \mathcal{T}_{\leq k}^{(2)}, \mathbf{p}^{(1)}, \mathbf{p}^{(2)}\right) = \widetilde{Q}_{k+1}\left(E_{k+1}^{r_{\leq k}^{(2)}}|\mathbf{p}^{(2)}, \widetilde{\mathcal{T}}_{\leq k}^{(2)}\right)$$

We now combine all the cases: let $\phi_{\leq k}^{(\mathsf{A})}, \phi_{\leq k}^{(\mathsf{R})}$ denote the conditional densities of $\widetilde{\mathcal{T}}_{\leq k}^{(2)}$ under events A and R respectively. Let $T_k = \left|\widetilde{\mathcal{T}}_{\leq k}^{(2)}\right|$. For any event $E$,

$$\mathbb{P}\left(\widetilde{\mathcal{T}}_{\leq k+1}^{(2)} \in E|\mathbf{p}^{(1)}, \mathbf{p}^{(2)}\right) = \mathbb{P}\left(r_{k+1}^{(2)} \in E_{k+1}^{r_{\leq k}^{(2)}}|\mathsf{A}, \widetilde{\mathcal{T}}_{\leq k}^{(2)} \in E_{\leq k}, \mathbf{p}^{(1)}, \mathbf{p}^{(2)}\right)\mathbb{P}\left(\widetilde{\mathcal{T}}_{\leq k}^{(2)} \in E_{k+1}^{r_{\leq k}^{(2)}}, \mathsf{A}|\mathbf{p}^{(1)}, \mathbf{p}^{(2)}\right)$$

$$+ \mathbb{P}\left(r_{k+1}^{(2)} \in E_{k+1}^{r_{\leq k}^{(2)}}|\mathsf{R}, \widetilde{\mathcal{T}}_{\leq k}^{(2)} \in E_{\leq k}, \mathbf{p}^{(1)}, \mathbf{p}^{(2)}\right)\mathbb{P}\left(\widetilde{\mathcal{T}}_{\leq k}^{(2)} \in E_{\leq k}, \mathsf{R}|\mathbf{p}^{(1)}, p^{(2)}\right)$$

$$= \int_{\mathbb{R}^{dT_{k+1}}} \mathbb{1}\left(r_{k+1}^{(2)} \in E_{k+1}^{r_{\leq k}^{(2)}}\right) \mathbb{1}\left(\widetilde{\mathcal{T}}_{\leq k}^{(2)} \in E_{\leq k}\right)\left(\mathbb{1}\left(\widetilde{\mathcal{T}}_{\leq k}^{(2)} \in \mathsf{A}\right) \phi_{\leq k}^{(\mathsf{A})}\left(\widetilde{\mathcal{T}}_{\leq k}^{(2)}\right)\right.$$

$$\left.+ \mathbb{1}\left(\widetilde{\mathcal{T}}_{\leq k}^{(2)} \in \mathsf{R}\right) \phi_{\leq k}^{(\mathsf{R})}\left(\widetilde{\mathcal{T}}_{\leq k}^{(2)}\right)\right)\phi_{\widetilde{Q}_{k+1}^{\mathbf{p}^{(2)}}}\left(r_{k+1}^{(2)}|\widetilde{\mathcal{T}}_{\leq k}^{(2)}\right) d\widetilde{\mathcal{T}}_{\leq k}^{(2)} dr_{k+1}^{(2)}$$

$$= \int_{\mathbb{R}^{dT_{k+1}}} \mathbb{1}\left(\mathcal{T}_{\leq k+1}^{(2)} \in E\right) \phi_{Q_{\leq k}^{p^{(2)}}}\left(\widetilde{\mathcal{T}}_{\leq k}^{(2)}\right) \phi_{\widetilde{Q}_{k+1}^{p^{(2)}}}\left(r_{k+1}^{(2)}|\widetilde{\mathcal{T}}_{\leq k}^{(2)}\right) d\widetilde{\mathcal{T}}_{\leq k}^{(2)} dr_{k+1}^{(2)}$$

$$= \widetilde{Q}_{\leq k+1}^{p^{(2)}}(E)$$

where in the third equality, we use the induction hypothesis. This completes the proof of the lemma. $\qquad\square$

**Lemma 5.** *For any measurable event $E \subseteq \mathfrak{T}$, $\mathbb{P}[\mathcal{T}^{(2)} \in E] = Q(E)$.*

*Proof.* This follows primarily from Lemma 4, and the fact that other elements in nodes of $\mathcal{T}$, namely $u_b$ and $w_b$ are deterministic functions of the prefix vertices in the tree $\widetilde{\mathcal{T}}$. Consider a decomposition of the event $E = E_u \times E_r \times E_w \times E_z$. Now,

$$
\begin{aligned}
\mathbb{P}[\mathcal{T}^{(2)} \in E] &= \mathbb{E}_{\mathbf{p}^{(1)}} \mathbb{P}\left(\mathcal{T}^{(2)} \in E_u \times E_r \times E_w \times E_z | \mathbf{p}^{(1)}, \mathbf{p}^{(2)} \in E_z\right) \mathbb{P}\left(\mathbf{p}^{(2)} \in E_z\right) \\
&= \mathbb{E}_{\mathbf{p}^{(1)}} \mathbb{P}\left(\widetilde{\mathcal{T}}^{(2)} \in E_r | \mathbf{p}^{(1)}, \mathbf{p}^{(2)}\right) \mu_{n-1}(E_z) \\
&= \mathbb{E}_{\mathbf{p}^{(1)}} \widetilde{Q}^{\mathbf{p}^{(2)}}(E_r) \mu_{n-1}(E_2) \\
&= \mathbb{E}_{\mathbf{p}^{(1)}} Q^{\mathbf{p}^{(2)}}(E_u \times E_w \times E_r) \mu_{n-1}(E_z) \\
&= Q(E)
\end{aligned}
$$

where the second and fourth equality follows since variables $w_b$ and $u_b$ are deterministic functions of the responses $r_{\leq b}$. The second and third equality also uses Lemma 3 and Lemma 4 respectively. $\qquad \square$

**Lemma 6.** *The probability of retraining is at most* $\log(n)\rho$.

*Proof.* A retraining is triggered only when a rejection sampling step fails. Note that a rejection sampling step happens only when the node $b$ belongs to the path from $s$ to root, say path. Let Accept be the event when all rejection sampling steps succeed.

$$
\begin{aligned}
\mathbb{P}(\text{Accept}) &= \mathbb{E}_{\mathcal{T}^{(1)}, \mathcal{T}^{(2)}, \{u_b\}} \prod_{b \in \text{path}} \mathbb{1}\left(u_b \leq \frac{\phi_{\widetilde{Q}_b^{\mathbf{p}^{(2)}}}\left(r_b^{(1)} | \mathcal{T}_{<b}^{(1)}\right)}{\phi_{\widetilde{P}_b^{\mathbf{p}^{(2)}}}\left(r_b^{(1)} | \mathcal{T}_{<b}^{(1)}\right)}\right) \\
&= \mathbb{E}_{\widetilde{\mathcal{T}}^{(1)}, \mathbf{p}^{(1)}, \mathbf{p}^{(2)}} \prod_{b \in \text{path}} \mathbb{P}\left(u_b \leq \frac{\phi_{\widetilde{Q}_b^{\mathbf{p}^{(2)}}}\left(r_b^{(1)} | \widetilde{\mathcal{T}}_{<b}^{(1)}\right)}{\phi_{\widetilde{P}_b^{\mathbf{p}^{(1)}}}\left(r_b^{(1)} | \widetilde{\mathcal{T}}_{<b}^{(1)}\right)}\right) \\
&= \mathbb{E}_{\mathbf{p}^{(1)}, \mathbf{p}^{(2)}} \prod_{b \in \text{path}} \int_{\mathbb{R}^d} \min\left(\phi_{\widetilde{Q}_b^{\mathbf{p}^{(2)}}}\left(r_b^{(1)} | \widetilde{\mathcal{T}}_{<b}^{(1)}\right), \phi_{\widetilde{P}_b^{\mathbf{p}^{(1)}}}\left(r_b^{(1)} | \widetilde{\mathcal{T}}_{<b}^{(1)}\right)\right) dr_b^{(2)} \\
&= \mathbb{E}_{\mathbf{p}^{(1)}, \mathbf{p}^{(2)}} \prod_{b \in \text{path}} \left(1 - \mathsf{TV}\left(\widetilde{Q}_b^{\mathbf{p}^{(2)}}, \widetilde{P}_b^{\mathbf{p}^{(1)}} | \widetilde{\mathcal{T}}_{<b}^{(1)}\right)\right) \\
&= \prod_{b \in \text{path}} (1 - \rho_b) \\
&\geq 1 - \sum_{b \in \text{path}} \rho_b \\
&\geq 1 - \log(n) \max_b \rho_b \\
&\geq 1 - \log(n)\rho
\end{aligned}
$$

where the fourth equality follows from the definition of TV distance and in the last equality, $\rho_b$ denotes the (conditional) TV distance of node $b$. The third to last inequality follows from Lemma 7 and the second to last inequality follows from Holder's inequality. For the last inequality, we simply upper bound $\rho_b \leq \rho$ since the algorithm is $\rho$-TV stable (Lemma 2). This completes the proof. $\qquad \square$

**Lemma 7.** *Let* $\{a_i\}_{i=1}^k$ *be real numbers such that* $a_i \in (0, 1)$ *for all* $i$ *and* $\sum_{i=1}^k a_i \leq 1$. *Then,* $\prod_{i=1}^k (1 - a_i) \geq 1 - \sum_{i=1}^k a_i$

*Proof.* We prove this via induction on $k$. The base case $k = 1$ is immediate. For the induction step $k$, we have

$$\prod_{i=1}^{k} (1 - a_i) = \prod_{i=1}^{k-1} (1 - a_i)(1 - a_k) \geq \left(1 - \sum_{i=1}^{k-1} a_i\right)(1 - a_k)$$

$$= 1 - \sum_{i=1}^{k} a_i + \left(\sum_{i=1}^{k-1} a_i\right) a_k$$

$$\geq 1 - \sum_{i=1}^{k} a_i$$

This completes the proof. $\square$

# E    MISSING PROOFS FROM SECTION 5

## E.1    VARIANCE-REDUCED FRANK WOLFE

---
**Algorithm 12** Variance-reduced Frank Wolfe$(t_0; \mathcal{T})$

---
**Input:** Dataset $S$, loss function $(w, z) \mapsto \ell(w, z)$, steps $T$, $\sigma$, $\{\eta_t\}_t$
1:  **if** $t_0 = 1$ **then** Permute dataset, initialize $\mathcal{T}$, set $w_{t_0} = 0$ **end if**
2:  **for** $t = 1$ to $T - 1$ **do**
3:     $u_t = \sum_{i=1}^{t} \left((i + 1)\nabla\ell(w_i; z_i) - i\nabla\ell(w_{i-1}; z_i)\right)$
4:     Append$(u_t, \sigma; \mathcal{T})$
5:     $r_t = $ GetPrefixSum$(t; \mathcal{T})$
6:     $v_t = \arg\min_{w \in \mathcal{W}} \left\langle w, \frac{r_t}{t+1} \right\rangle$
7:     $w_{t+1} = (1 - \eta_t)w_t + \eta_t v_t$
8:     Set$(\text{leaf}(t), (u_t, r_t, w_t, z_t); \mathcal{T})$
9:  **end for**
**Output:** $\widehat{w} = w_T$

---

*Proof of Theorem 2.* For the accuracy guarantee, we follow the proof of Theorem 1 in Zhang et al. (2020). Let $d_t = \frac{r_t}{t+1}$. From smoothness, we have

$$L(w_{t+1}; \mathcal{D}) \leq L(w_t; \mathcal{D}) + \langle \nabla L(w_t; \mathcal{D}), w_{t+1} - w_t \rangle + \frac{H}{2} \|w_{t+1} - w_t\|^2$$

$$\leq L(w_t; \mathcal{D}) + \eta_t \langle \nabla L(w_t; \mathcal{D}) - d_t, v_t - w_t \rangle + \langle d_t, v_t - w_t \rangle + \frac{\eta_t^2 H D^2}{2}$$

$$= L(w_t; \mathcal{D}) + \eta_t \langle \nabla L(w_t; \mathcal{D}) - d_t, v_t - w_t \rangle + \eta_t \langle d_t, w^* - w_t \rangle + \frac{\eta_t^2 H D^2}{2}$$

$$\leq L(w_t; \mathcal{D}) + \eta_t \langle \nabla L(w_t; \mathcal{D}), w^* - w_t \rangle + \eta_t \langle d_t - \nabla L(w_t), w^* - v_t \rangle + \frac{\eta_t^2 H D^2}{2}$$

$$\leq (1 - \eta_t)L(w_t; \mathcal{D}) - \eta_t L(w^*; \mathcal{D}) + \frac{2D}{t+1} \|d_t - \nabla L(w_t; \mathcal{D})\| + \frac{\eta_t^2 H D^2}{2}$$

where the second inequality follows from the update and the fact that iterates lie in the set of diameter $D$. The third inequality follows from the optimality of $v_t$ in the update in Algorithm 12. Finally, the last inequality follows from convexity, Cauchy-Schwarz inequality and by substituting the step-size. We now take expectation, and use the bound on gradient estimation error in Lemma 8 to get,

$$\mathbb{E}[L(w_{t+1}; \mathcal{D}) - L(w^*; \mathcal{D})]$$

$$\leq (1 - \eta_t)\mathbb{E}[L(w_t; \mathcal{D}) - L(w^*; \mathcal{D})] + \widetilde{O}\left((HD + G)D\left(\frac{1}{(t+1)^{3/2}} + \frac{\sqrt{d}}{(t+1)^2 \rho}\right)\right)$$

$$+ \frac{HD^2}{2(t+1)^2}$$

The above recursion gives us,

$$\mathbb{E}[L(w_T; \mathcal{D}) - L(w^*; \mathcal{D})] \leq (L(w_1; \mathcal{D}) - L(w^*)) \prod_{t=1}^{T-1} (1 - \eta_t)$$

$$+ \sum_{i=1}^{T-1} \widetilde{O}\left((HD + G)\, D\left(\frac{1}{(i+1)^{3/2}} + \frac{\sqrt{d}}{(i+1)^2\,\rho}\right) + \frac{HD^2}{(i+1)^2}\right) \prod_{t=i+1}^{T-1} (1 - \eta_t)$$

$$\leq \frac{HD^2}{T}$$

$$+ \sum_{i=1}^{T-1} \widetilde{O}\left((HD + G)\, D\left(\frac{1}{(i+1)^{1/2}} + \frac{\sqrt{d}}{(i+1)\,\rho}\right) + \frac{HD^2}{(i+1)}\right) \frac{1}{T}$$

$$\leq \widetilde{O}\left((HD + G)\, D\left(\frac{1}{\sqrt{T}} + \frac{\sqrt{d}}{T\rho}\right) + \frac{HD^2}{T}\right)$$

$$\leq \widetilde{O}\left((HD + G)\, D\left(\frac{1}{\sqrt{T}} + \frac{\sqrt{d}}{T\rho}\right)\right)$$

where the second inequality follows from smoothness and substituting $\prod_{t=i+1}^{T-1}(1 - \eta_t) = \frac{i+1}{T-1}$. Substituting number of iterations $T = n$ completes the accuracy proof.

For the unlearning part, we start by showing that the algorithm falls into the template of bounded sensitivity prefix-sum query release. Recall that the update $u_t = \sum_{i=1}^{t}\left((i+1)\,\nabla\ell(w_i; z_i) - i\nabla\ell(w_{i-1}; z_i)\right)$.

The sensitivity is then bounded as,

$$\|((i+1)\,\nabla\ell(w_i; z) - i\nabla\ell(w_{i-1}; z)) - ((i+1)\,\nabla\ell(w_i; z') - i\nabla\ell(w_{i-1}; z'))\|$$
$$\leq iH\,\|w_i - w_{i-i}\| + 2G$$
$$\leq iH\eta_{i-1}\,\|v_{i-1} - w_{i-1}\| + 2G$$
$$\leq 2\,(HD + G)$$

where the first inequality follows from smoothness and Lipschitzness of the loss. The second inequality follows from the update in Algorithm 12 and the last inequality follows from the fact that the iterates remain in the set of diameter $D$. Hence the correctness of the unlearning algorithm follows from Theorem 1. For runtime, the training time, in terms of gradient computations is $\Theta(n)$. Therefor, using the fact that the relative unlearning complexity, from Theorem 1, is $\widetilde{O}(\rho)$, we have $\widetilde{O}(\rho n)$ bound on expected unlearning runtime. $\qquad\square$

**Lemma 8.** *The gradient estimation error* $\mathbb{E}\left\|\frac{r_t}{t+1} - \nabla L(w_t; \mathcal{D})\right\|^2 \leq \widetilde{O}\left((HD + G)^2\left(\frac{1}{t+1} + \frac{d}{(t+1)^2\rho^2}\right)\right)$

*Proof.* Note that $d_t := \frac{r_t}{t+1}$ comprises of the original gradient estimate from Zhang et al. (2020), say $\widetilde{d}_t$ and the noise added by the binary tree mechanism, say $\xi_t$. Hence,

$$\mathbb{E}\,\|d_t - \nabla L(w_t; \mathrm{D})\|^2 = \mathbb{E}\left\|\widetilde{d}_t - \nabla L(w_t; \mathrm{D})\right\|^2 + \mathbb{E}\,\|\xi_t\|^2$$

$$\leq \widetilde{O}\left(\frac{(HD + G)^2}{t+1}\right) + \sum_{i=1}^{\log(n)} \frac{d\sigma^2}{(t+1)^2\,\rho^2}$$

$$= \widetilde{O}\left((HD + G)^2\left(\frac{1}{t+1} + \frac{d}{(t+1)^2\,\rho^2}\right)\right)$$

where the first inequality follows from Lemma 2 in Zhang et al. (2020) with $\alpha = 1$, and the fact that in the binary tree mechanism we add noise of variance $\sigma$ at most $\log(n)$ times; the factor $1/(t+1)^2$

comes because the gradient estimate is $r_t/(t+1)$ and $r_t$ is the binary tree response. The final equality follows by plugging in the value of $\sigma$. □

### E.2 DUAL AVERAGING

---

**Algorithm 13** Dual averaging$(t_0; \mathcal{T})$

---

**Input:** Dataset $S$, loss function $(w, z) \mapsto \ell(w, z)$, steps $T$, $\{\eta_t\}_t$,
1: **if** $t_0 = 1$ **then** Permute dataset, initialize $\mathcal{T}$, set $w_{t_0} = 0$ **end if**
2: **for** $t = 1$ to $T - 1$ **do**
3:   $u_t = \sum_{i=1}^t \nabla \ell(w_i; z_i)$
4:   Append$(u_t, \sigma; \mathcal{T})$
5:   $r_t = \mathsf{GetPrefixSum}(t; \mathcal{T})$
6:   $w_{t+1} = \Pi_{\mathcal{W}} (w_0 - \eta_t p_t)$
7:   Set$(\mathsf{leaf}(t), (u_t, r_t, w_t, z_t); \mathcal{T})$
8: **end for**
**Output:** $\widehat{w} = w_T$

---

*Proof of Theorem 3.* The accuracy guarantee directly follows from Theorem 5.1 in Kairouz et al. (2021), replacing $\epsilon/\log^2(1/\delta)^2$ therein by $\rho$. To elaborate, we set $\sigma = \widetilde{O}\left(\frac{G^2}{\rho^2}\right)$ as opposed to $\widetilde{O}\left(\frac{G^2 \log^4(1/\delta)}{\epsilon^2}\right)$, hence substituting it in the accuracy proof of Theorem 5.1 in Kairouz et al. (2021) gives the claimed guarantee.

For the unlearning part, we start by showing that the algorithm falls into the template of bounded sensitivity prefix query release.

Recall that the update $u_t = \sum_{i=1}^t \nabla \ell(w_t; z_i)$. The sensitivity is simply bounded by Lipschitznes as,

$$\|\nabla \ell(w_t; z) - \nabla \ell(w_t; z')\| \leq 2G$$

Hence the correctness of the unlearning algorithm follows from Theorem 1. For runtime, the training time, in terms of gradient computations is $\Theta(n)$. Therefor, using the fact that the relative unlearning complexity, from Theorem 1, is $\widetilde{O}(\rho)$, we have $\widetilde{O}(\rho n)$ bound on expected unlearning runtime. □

### E.3 CONVEX GLMS WITH THE JL METHOD

**Theorem 5.** *Let $\rho \leq 1$ and $\ell : \mathcal{W} \times \mathcal{X} \times \mathcal{Y} \to \mathbb{R}$ be an $H$-smooth, $G$-Lipschitz convex GLM loss function. Algorithm 4 instantiated with Algorithm 12, as the learning algorithm, run with $\sigma^2 = \widetilde{O}\left(\frac{(H\|\mathcal{X}\|^2\|w^*\| + G\|\mathcal{X}\|)^2}{\rho^2}\right)$, $t_0 = 1$, $\eta_t = \frac{1}{t+1}$ and $k = \widetilde{O}\left(\left(\frac{H\|\mathcal{X}\|^2\|w^*\|}{(H\|\mathcal{X}\|^2\|w^*\| + G\|\mathcal{X}\|)}\right)^{2/3}(n\rho)^{2/3}\right)$ on a dataset $S$ of $n$ samples, drawn i.i.d. from $\mathcal{D}$, outputs $\widehat{w}$ with excess population risk bounded as*
$$\mathbb{E}\left[L(\widehat{w}; \mathcal{D}) - L(w^*; \mathcal{D})\right] = \widetilde{O}\left(\frac{(G\|\mathcal{X}\| + H\|\mathcal{X}\|^2\|w^*\|)\|w^*\|}{\sqrt{n}} + \frac{H^{1/3}G^{2/3}\|w^*\|^{4/3}\|\mathcal{X}\|^{4/3} + H\|\mathcal{X}\|^2\|w^*\|^2}{(n\rho)^{2/3}}\right)$$

*Furthermore, the corresponding unlearning Algorithm 3 (with query and update functions as specified in the learning algorithm), satisfies exact unlearning with $\widetilde{O}(\rho n)$ expected runtime .*

*Proof of Theorem 5.* We start with the accuracy guarantee. Let $\alpha \leq 1$ be a parameter to be set later. From the JL property, with $k = O\left(\log(n/\beta)/\alpha^2\right)$, with probability at least $1 - \beta$, the norm of all data-points in $S$, $\|\Phi x_i\| \leq (1 + \alpha)\|\mathrm{x}_i\| \leq 2\|\mathcal{X}\|$. Hence, conditioned on the above event, the GLM loss function function is $\widetilde{G} = 2G\|\mathcal{X}\|$-Lipschitz and $\widetilde{H} = 4H\|\mathcal{X}\|^2$-smooth. Let $\Phi\mathcal{D}$ denote the push-forward measure of $\mathcal{D}$ under the map $(x, y) \mapsto (\Phi x, y)$. With probability at least $1 - \beta$, the excess risk is,

$$
\begin{aligned}
\mathbb{E}[L(\widehat{w};\mathcal{D}) - L(w^*;\mathcal{D})] &= \mathbb{E}[L(\Phi^\top \widetilde{w};\mathcal{D}) - L(\Phi w^*;\Phi\mathcal{D})] + \mathbb{E}[L(\Phi w^*;\Phi\mathcal{D}) - L(w^*;\mathcal{D})] \\
&= \mathbb{E}[L(\widetilde{w};\Phi\mathcal{D}) - L(\Phi w^*;\Phi\mathcal{D})] + \mathbb{E}[\phi_y(\langle \Phi w^*, \Phi x\rangle) - \phi_y(\langle w^*, x\rangle)] \\
&\leq \widetilde{O}\left(\left(\widetilde{G} + \widetilde{H}\,\|w^*\|\right)\|w^*\|\left(\frac{1}{\sqrt{n}} + \frac{\sqrt{k}}{n\rho}\right)\right) + \frac{H}{2}\mathbb{E}\,|\langle \Phi x, \Phi w^*\rangle - \langle x, w^*\rangle|^2 \\
&\leq \widetilde{O}\left(\left(\widetilde{G} + \widetilde{H}\,\|w^*\|\right)\|w^*\|\left(\frac{1}{\sqrt{n}} + \frac{\sqrt{k}}{n\rho}\right) + \frac{\widetilde{H}\,\|w^*\|^2}{k}\right) \\
&= \widetilde{O}\left(\frac{\left(\widetilde{G} + \widetilde{H}\,\|w^*\|\right)\|w^*\|}{\sqrt{n}} + \frac{\widetilde{H}^{1/3}\widetilde{G}^{2/3}\,\|w^*\|^{4/3} + \widetilde{H}\,\|w^*\|^2}{(n\rho)^{2/3}}\right)
\end{aligned}
$$

where in the first inequality, we use the accuracy guarantee of VR-Frank Wolfe (Theorem 2) and smoothness of $\phi_y$ together with the fact that $w^*$ is globally optimal. The second inequality follows from JL property and the last inequality follows by the setting of $k$.

For the in-expectation (over the JL matrix) bound, note that in the worst-case, $L(\widehat{w};\mathcal{D}) - L(w^*;\mathcal{D}) \leq G\,\|\widehat{w} - w^*\|$. From boundedness of the range of (typical) JL maps, $\|\widehat{w} - w^*\| = \text{poly}(n,d)$ w.p. 1. Hence, taking the failure probability $\beta$ to be small enough suffices to be give an expectation bound which is same as above upto polylogarithmic factors.

We now proceed to the unlearning guarantee. We first remark that the correctness of the unlearning algorithm (see Lemma 4) holds as long as the learning algorithm uses prefix-sum queries, even with *unbounded* sensitivity. Hence, the correctness follows. We now proceed to bound the unlearning runtime. We first bound the TV stability parameter of the learning algorithm using Lemma 9. The setting of noise variance $\sigma$ in Algorithm 4 together with the stability guarantee of Theorem 2 ensures that $\gamma(\widetilde{H},\widetilde{G}) \leq \frac{\tau}{2}$. Hence the JL method satisfies $\rho$-TV stability. Now, Lemma 6 gives us that the probability of retraining is at most $\widetilde{O}(\rho)$. Since the training time, in terms of gradient computations is $\Theta(n)$, we have $\widetilde{O}(\rho n)$ bound on expected unlearning runtime. $\qquad\square$

**Theorem 6.** *Let $\rho \leq 1$ and $\ell : \mathcal{W} \times \mathcal{X} \times \mathcal{Y} \to \mathbb{R}$ be a G-Lipschitz convex GLM loss function. Algorithm 4 with Algorithm 13 as the sub-routine, as the learning algorithm, run with $\sigma^2 = O\left(\frac{G^2\|\mathcal{X}\|^2}{\rho^2}\right)$, $t_0 = 1$, $\eta = \frac{\|w^*\|d^{1/4}\sqrt{\log(n)}}{G\|\mathcal{X}\|\sqrt{n\rho}}$ and $k = \sqrt{n\rho}$ on a dataset $S$ of $n$ samples sampled i.i.d. from $\mathcal{D}$ outputs $\widehat{w}$, with excess population risk bounded as, $\mathbb{E}\left[L(\widehat{w};\mathcal{D}) - L(w^*;\mathcal{D})\right] = \widetilde{O}\left(G\|\mathcal{X}\|\,\|w^*\|\left(\frac{1}{\sqrt{n}} + \frac{1}{(n\rho)^{1/3}}\right)\right)$. Furthermore, the corresponding unlearning Algorithm 3 (with query and update functions as specified in the learning algorithm), satisfies exact unlearning with $\widetilde{O}(\rho n)$ expected runtime.*

*Proof of Theorem 6.* We start with the accuracy guarantee; let $\alpha \leq 1$ be a parameter to be set later. From the JL property, with $k = O\left(\log(n/\beta)/\alpha^2\right)$, with probability at least $1 - \beta$, the norm of all data-points in $S$, $\|\Phi x_i\| \leq (1 + \alpha)\|x_i\| \leq 2\|\mathcal{X}\|$. Hence, conditioned on the above event, the GLM loss function function is $\widetilde{G} = 2G\|\mathcal{X}\|$-Lipschitz. Let $\Phi\mathcal{D}$ denote the push-forward measure of $\mathcal{D}$ under the map $(x,y) \mapsto (\Phi x, y)$. With probability at least $1 - \beta$, the excess risk is,

$$
\begin{aligned}
\mathbb{E}[L(\widehat{w};\mathcal{D}) - L(w^*;\mathcal{D})] &= \mathbb{E}[L(\Phi^\top \widetilde{w};\mathcal{D}) - L(\Phi w^*;\Phi\mathcal{D})] + \mathbb{E}[L(\Phi w^*;\Phi\mathcal{D}) - L(w^*;\mathcal{D})] \\
&= \mathbb{E}[L(\widetilde{w};\Phi\mathcal{D}) - L(\Phi w^*;\Phi\mathcal{D})] + \mathbb{E}[\phi_y(\langle \Phi w^*,\Phi x\rangle) - \phi_y(\langle w^*,x\rangle)] \\
&\leq \widetilde{O}\left(\widetilde{G}\,\|w^*\|\left(\frac{1}{\sqrt{n}} + \sqrt{\frac{\sqrt{k}}{n\rho}}\right)\right) + G\mathbb{E}\,|\langle \Phi x,\Phi w^*\rangle - \langle x,w^*\rangle| \\
&\leq \widetilde{O}\left(\widetilde{G}\,\|w^*\|\left(\frac{1}{\sqrt{n}} + \sqrt{\frac{\sqrt{k}}{n\rho}}\right) + \frac{\widetilde{G}\,\|w^*\|}{\sqrt{k}}\right) \\
&\leq \widetilde{O}\left(\widetilde{G}\,\|w^*\|\left(\frac{1}{\sqrt{n}} + \frac{1}{(n\rho)^{1/3}}\right)\right)
\end{aligned}
$$

where in the first inequality, we use the accuracy guarantee of Dual Averaging (Theorem 3) and Lipschitzness of $\phi_y$ together. The second inequality follows from JL property and the last inequality follows by the setting of $k$. As in Theorem 5, the same bound as above for in-expectation (over the JL matrix) holds follows by taking the failure probability $\beta$ to be small enough.

The correctness and runtime of the unlearning algorithm follows as in the proof of Theorem 5. $\qquad\square$

**Lemma 9.** *Suppose $\mathcal{A}$ is an algorithm which when run on $\widetilde{H}$-smooth and $\widetilde{G}$-Lipschitz functions is $\gamma(\widetilde{H},\widetilde{G})$-TV stable, then the JL method with with $k = O\left(\log\left(2n/\tau\right)\right)$ and $\mathcal{A}$ as input, run on $H$-smooth and $G$-Lipschitz GLMs, satisfies $\frac{\tau}{2} + \gamma\left(2G\,\|\mathcal{X}\|, 4H\,\|\mathcal{X}\|^2\right)$-TV stability.*

*Proof.* Given a dataset $S$ let $G_S$ be the uniform bound on Lipschitzness parameter of the class of loss functions $\{w \mapsto \ell(w;z)\}_{z\in S}$. We define $H_S$ similarly. Let $\alpha \leq 1$ be a parameter to be set later. From the JL property, with $k = O\left(\log\left(n/\beta\right)\right)$, with probability at least $1 - \beta$, the norm of all data-points in $S$, $\|\Phi x_i\| \leq 2\,\|\mathcal{X}\|$ - we denote this event as $E_{\mathrm{JL}}$. Since the loss function is a GLM, we have that conditioned on $E_{\mathrm{JL}}$, the Lipschitzness and smoothness parameters $G_S$ and $H_S$ are bounded by $2G\,\|\mathcal{X}\|$ and $2H\,\|\mathcal{X}\|^2$ respectively. We therefore get a stability parameter $\widetilde{\gamma} := \gamma\left(2G\,\|\mathcal{X}\|, 4H\,\|\mathcal{X}\|^2\right)$.

We set $\beta = \rho/2$. We now incorporate the failure probability in the failure guarantee. Let $P_\Phi$ and $Q_\Phi$ denote the probability distributions of the output on datasets $S$ and $S'$. By definition of TV distance,

$$
\begin{aligned}
\mathrm{TV}(P_\Phi, Q_\Phi) &= \sup_E \mathbb{P}_{w\sim P}\left(w\in E\right) - \mathbb{P}_{w\sim Q}\left(w\in E\right) \\
&= \sup_E \Big(\mathbb{P}_{w\sim P}\left(w\in E|E_{\mathrm{JL}}\right)\mathbb{P}(E_{\mathrm{JL}}) + \mathbb{P}_{w\sim P}\left(w\in E|E'_{\mathrm{JL}}\right)\mathbb{P}(E'_{\mathrm{JL}}) \\
&\quad - \mathbb{P}_{w\sim Q}\left(w\in E|E_{\mathrm{JL}}\right)\mathbb{P}(E_{\mathrm{JL}}) - \mathbb{P}_{w\sim Q}\left(w\in E|E'_{\mathrm{JL}}\right)\mathbb{P}(E'_{\mathrm{JL}})\Big) \\
&\leq \left(\sup_E \mathbb{P}_{w\sim P}\left(w\in E|E_{\mathrm{JL}}\right) - \mathbb{P}_{w\sim Q}\left(w\in E|E_{\mathrm{JL}}\right)\right)\mathbb{P}(E_{\mathrm{JL}}) \\
&\quad + \left(\sup_E \mathbb{P}_{w\sim P}\left(w\in E|E'_{\mathrm{JL}}\right) - \mathbb{P}_{w\sim Q}\left(w\in E|E'_{\mathrm{JL}}\right)\right)\mathbb{P}(E'_{\mathrm{JL}}) \\
&\leq \left(\sup_E \mathbb{P}_{w\sim P}\left(w\in E|E_{\mathrm{JL}}\right) - \mathbb{P}_{w\sim Q}\left(w\in E|E_{\mathrm{JL}}\right)\right) + \rho/2 \\
&\leq \widetilde{\gamma} + \rho/2
\end{aligned}
$$

which completes the proof. $\qquad\square$

## F SCO IN DYNAMIC STREAMS

In this section, we extend our previous results to dynamic streams wherein we observe a sequence of insertions and deletions, starting with potentially zero data points. We assume that the number of

available points throughout is positive and the data points are i.i.d. from an an unknown distribution as well as the unlearning requests are chosen independent of the algorithm.

To give a simple and unified presentation, let the expected accuracy, say excess population risk, of the $\rho$-TV stable Algorithm 2 with a dataset $S$ be $\alpha(|S|, \rho; \mathcal{P})$ where $\mathcal{P}$ denotes problem specific parameters such as Lipschitzness, diameter etc.

We present two techniques for dynamic streams; the first one satisfies exact unlearning but has a worse update time; this is similar to Ullah et al. (2021). The other satisfies weak unlearning (defined below) with better update time. A key component to both are *anytime* guarantees described below.

**Anytime binary tree mechanism:** In the previous section, the depth of the initialized tree and the noise variance $\sigma$, both were chosen as a function of the dataset size $n$. However, the tree can be easily built in an online manner as in prior work Guha Thakurta & Smith (2013). For setting the noise variance: for target $\rho$-TV stability, we distribute the noise budget exponentially along the height of the tree; specifically, the leaf node contribute to $\rho/2$ TV stability, the nodes above them $\rho/4$ and so on. In this way, the final tree satisfies $\rho$-TV stability for any value of $n$.

**Anytime accuracy:** The other problem of changing data size is that the internal parameters of algorithm (step size, in our case) may be set as a function of $n$ for desirable accuracy guarantees. Fortunately, the two algorithms that we consider, VR-Frank Wolfe and Dual Averaging, have known horizon-oblivious parameter settings Orabona (2019). Their JL counterparts on the other hand, require setting the embedding dimension as a function of $n$, and thus not applicable unless we assume that the number of data points throughout the stream is $\Theta(n)$.

### F.1 Weak Unlearning

We first define weak unlearning wherein only the model output and not the entire state is required to be indistinguishable.

**Definition 9** (Weak unlearning). *A procedure* $(\mathbf{A}, \mathbf{U})$ *satisfies weak unlearning if for all all datasets* $S$, *all* $z \in \mathcal{Z}$, *and for all events* $\mathcal{E} \subseteq \mathcal{W} \times \mathcal{M}$, *we have,* $\mathbb{P}(\mathcal{A}(S \setminus \{z\}) \in \mathcal{E}) = \mathbb{P}(\mathcal{U}(\mathbf{A}(S), z) \in \mathcal{E})$

We now argue in what way insertions handled in Ullah et al. (2021) is deficient. The main reason is that they require insertions to also satisfy the unlearning criterion: the state of the system upon insertion is instinguishable to the state had the inserted point being present to begin with. However, this is an overkill; adding new points simply serve to yield improved statistical accuracy. Furthermore, methods which allow adding new points, abound, particularly in the stochastic optimization setting, popularly known as *incremental* or continual release methods. Importantly, the insertion time of these methods is constant (in $n$). Hence, a natural question is whether, for dynamic streams, can we design unlearning methods in which we pay for update time only in proportion to the number of deletions? Our result shows that we can, albeit under the weak unlearning (see Definition 9) guarantee.

Specifically, our procedure requires *hiding* the order in which data points are processed. Intuitively, an incremental method typically processes the newest data point the last. This ordering is problematic to the unlearning guarantee, since if some point is to deleted, then we can no longer replace it with the last point as that would result in a different order. Our main result is as follows.

**Theorem 7.** *In the dynamic streaming setting with* $R$ *requests, using anytime incremental learning and unlearning algorithms, Algorithm 2 and 3, without permuting the dataset, the following are true.*

1. *It satisfies weak unlearning at every time point in the stream.*

2. *The accuracy of the output* $\widehat{w}_i$ *at time point* $i$, *with corresponding dataset* $S_i$, *is*
$$\mathbb{E}[L(\widehat{w}_i; \mathcal{D})] - \min_w L(w; \mathcal{D}) = \alpha(\rho, |S_i|; \mathcal{P})$$

3. *The number of times retraining is triggered, for* $V$ *unlearning requests is at most* $\widetilde{O}(\rho V)$

Importantly, in the above guarantee, we only pay for the number of unlearning requests $V$ rather than the number of requests $R$.

*Proof of Theorem 7.* The first claim, weak unlearning guarantee of the unlearning algorithm, follows mainly from Lemma 4. Specifically, it shows that conditioned on the permutation of the dataset (in this case, since the dataset is not permuted, the permutation is simply identity), the distribution over

the responses $(r_b)_b$ in the tree after unlearning, is transported to the distribution of the output under $S'$. Since the model output is a deterministic function of the responses, (weak unlearning) correctness follows for one request. For the streaming setting, we simply apply the above inductively over the requests.

The second claim follows since, at every time point, the executed algorithm is indistinguishable from the base algorithm executed over the current dataset. Moreover, by assumption, the base algorithm, is *anytime*, i.e. no parameter is set which depends on the size of the dataset. Hence, the accuracy guarantee follows. For the last claim about the number of retraining, firstly, as motivated, by the assumption that the algorithm is incremental, the insertions are handled in $O(1)$ time. For the unlearning requests, note that from $\rho$-TV stability at every point, using Lemma 6, we have a $\widetilde{O}(\rho)$ probability of retraining. We now apply Proposition 8 from Ullah et al. (2021) which converts this to a bound on the expected number of times a retraining is triggered. For $V$ unlearning requests, this gives us a $\widetilde{O}(\rho V)$ bound on the number of retraining triggers. □

### F.2 EXACT UNLEARNING

Another way to extend the results for one unlearning request to dynamic streams is to modify the definition of unlearning (Definition 1) to also hold for insertions, as is done in Ullah et al. (2021). This allows us to apply the same tree based unlearning technique when handing insertions. Specifically, upon inserting a new point, we randomly choose a leaf and replace the leaf with the inserted point, and then insert the chosen leaf as the last leaf in the tree. We have the following guarantee for this method.

**Theorem 8.** *In the dynamic streaming setting with $R$ requests, using anytime learning and unlearning algorithms, Algorithm 2 and 3, the following are true.*

1. *Exact unlearning at every time point in the stream.*

2. *The accuracy of the output $\widehat{w}_i$ at time point $i$, with corresponding dataset $S_i$, is*
$$\mathbb{E}[L(\widehat{w}_i; \mathcal{D})] - \min_w L(w; \mathcal{D}) = \alpha(\rho, |S_i|; \mathcal{P})$$

3. *The total number of times, a retraining is triggered, for $R$ requests is at most $O(\rho R)$*

*Proof.* The arguments are similar to that of the proof of Theorem 7. The first part follows by applying the correctness of the unlearning algorithm, Theorem 1, inductively over the stream. We remark that the handling the insertions in the same way as deletions hardly changes anything in the proofs. The second claim follows from the anytime nature of the algorithm and by assumption on the accuracy guarantee. Finally, using the probability of retraining in Lemma 6 and Proposition 8 in Ullah et al. (2021) gives us the stated number of retraining triggers. □

