# OpenReview forum: "From Adaptive Query Release to Machine Unlearning"
_ICLR.cc/2023/Conference — Submitted to ICLR 2023_

### Official Review · Reviewer_44f4 · 2022-10-22

**Confidence:** 3
**Clarity, Quality, Novelty And Reproducibility:** The prefix-sum idea seems to be new i…
**Correctness:** 4
**Technical Novelty And Significance:** 2
**Empirical Novelty And Significance:** 2
**Recommendation:** 6

**Strength And Weaknesses:**

Strength. The paper provide improved oracle complexity for stochastic convex optimization (SCO) of the machine unlearning task. The prefix-sum idea seems simple, but it is good enough for one to use powerful tool from SCO. Meanwhile, I expect the prefix-sum subroutine could be useful for other applications of machine unlearning


Weakness. There is no major weakness.

**Summary Of The Paper:**

The paper studies the problem of machine unlearning and improves oracle complexity for stochastic convex optimization (SCO) under both smooth and non-smooth setting. The key idea is a new prefix-sum unlearning sub-routine that enables one to use Variance reduction Frank-wolfe and dual averaging algorithm for SCO.

Result. For smooth SCO, the paper gives an $O(\frac{1}{\sqrt{n}} + \frac{\sqrt{d}}{n\rho})$ accuracy answer with expected runtime $O(\rho n)$ per update using variance reduced Frank Wolfe. For non-smooth SCO, it gives $O(\frac{1}{\sqrt{n}}) + \frac{d^{1/4}}{\sqrt{n\rho}})$ accurate answer with $O(\rho n)$ per update, using dual averaging. Dimension independent results have also been obtained for the special case of generalized linear model.

**Summary Of The Review:**

The paper gives improved rate for stochastic convex optimization of machine unlearning with some interesting ideas. I would vote for acceptance.

Question for authors. Are there any lower bounds known for SCO of machine unlearning?


-----------

Post rebuttal

I have read the rebuttal and my evaluation remains.

---

> ### Author Response · Authors · 2022-11-11
> **Respone to Reviewer 44f4**
>
> We thank the reviewer for their feedback and suggestions. We note that we have revised the paper to incorporate the reviwers' comments (see text in red in the revision).
>
> **Are there any lower bounds known for SCO of machine unlearning?**
>
> As of know, there are no known lower bounds. Moreover, lower bounds would require more careful formalization of the problem, in terms of what is the instance space, what is the precise restriction on the algorithm and the information complexity model. We think that establishing non-trivial lower bounds may require significant work.

---

### Official Review · Reviewer_qNGF · 2022-10-23

**Confidence:** 3
**Correctness:** 3
**Technical Novelty And Significance:** 3
**Empirical Novelty And Significance:** Not applicable
**Recommendation:** 6

**Clarity, Quality, Novelty And Reproducibility:**

The main technical contribution of the work is novel and original. In terms of clairty, I believe the paper requires more polishing to be readable.

**Strength And Weaknesses:**

The paper claims a set of efficient algorithms for exact machine unlearning of SCO methods that are widely used in ML practice.  I believe this is a well-motivated question and the technical contributions here are novel and non-trivial.

However, I am concerned about the technical clarity of the presentation. Overall, I find the paper lack of some conceptual exposition to be accessible to non-experts of the area. In fact, much of the materials rely upon using the prior work Ullah et al. (2021). I believe for those who are not familiar with Ullah et al. (2021), the technical sections of the paper would be hard to parse.

Most importantly, the main algorithm from Section 4 (on prefix sum queries) is mostly described by the pseudocode. I find it hard to understand it at a conceptual level. Why and how does it perform unlearning? What does the noise-adding mechanisms do here? I think the issue stems from not having sufficient exposition on the coupling nature of the problem and the technique of reflection coupling.  (It appears that the techniques are borrowed from Ullah et al. (2021), which I have not read.)  I suggest the author(s) add a technical overview section earlier in the paper.

The paper does not give empirical validation of their methods, though I do not consider it a major weakness, as this work is theoretical in nature.

Some other minor comments:
----
In what sense is “linear queries” (Definition 8) is “linear in the data-points”, as claimed in the introduction? In Definition 8, the function p_j can be non-linear in z_j. In fact, if p_j is the gradient of the loss function, it can easily be non-linear in z_j (for many ML models and choices of the loss function).

After Definition 6 it should be mentioned that the bounded sensitivity condition is satisfied in variety of applications.

I think Appendix B requires more exposition. What does Algorithm 6 do at an intuitive level?  What is the LearnLinearQueries function? In fact, I suggest the author(s) expand this section and put it into the main paper as a warm up before delving into the prefix sum queries.

The paper should also discuss related work in more detail. In addition to Ullah et al. (2021) that this work builds upon, there’s been a flurry of recent work on machine unlearning. This includes, for example, Descent-to-Delete: Gradient-Based Methods for Machine Unlearning by Seth Neel, Aaron Roth, Saeed Sharifi-Malvajerdi (ALT 21)

Mechanical suggestions
---
1. In the additional prelim section, I suggest the author(s) to elaborate on reflection coupling, especially its formal guarantees. (The original paper claims the result in the setting of Brownian motion, which seems different from what’s going on here.)
2. End of page 2: Frank Wolfe -> “Frank-Wolfe”
3. Section 5: “[...], and thus fits into the framework”
4. Algorithm 6, the Load() function is undefined?


**Summary Of The Paper:**

The paper provides a framework for efficient machine unlearning for stochastic convex optimization. In particular, the work builds upon the prior paper by Ullah et al. (2021) and generalize it to handle prefix sum queries. As for concrete applications, this enables machine unlearning for variance-reduced  stochastic convex optimization (SCO) methods and dual averaging schemes.

**Summary Of The Review:**

Given the quality of the presentation, I cannot recommend accept. However, I can see that the paper could be improved a lot if the author(s) put some efforts on writing during the rebuttal period.

---

> ### Author Response · Authors · 2022-11-11
> **Response to Reviwer qNGF (part 1)**
>
> We thank the reviewer for their feedback. We have corrected the typos pointed out and incorporated the reviewer’s suggestions in the revised version (see text in red). Even though the reviewer acknowledges that our contributions are novel and non-trivial, unfortunately, we think that their score of 3 is too harsh and unjustified.
>
> **Sufficient exposition on the coupling nature of the problem**
>
> Contrary to what the reviewer says, we indeed motivate the coupling nature of the problem: see discussion after definition 2, wherein we define what a maximal coupling is and how it is relevant to the unlearning problem. We have now added more details in this part which hopefully helps understand the framework better (see text in red in the revision). We think that this, together with reflection coupling, which we describe (also, see below) is all there about couplings to know to understand everything in our paper.
>
> **What does the noise-adding mechanism do here?**
>
> To give some intuition, consider some task, eg; mean computation, and think about the two extremes: (a). no noise and (b) large (infinite) noise. In the first case, the output is most accurate, but requires (potentially) full retraining during unlearning. In the second case, the output has trivial accuracy, but requires no work during unlearning, since the output is just noise regardless of the dataset. Adding controlled noise allows one to trade-off accuracy with unlearning complexity.
>
>
> We model this intermediate point by the Total Variation (TV) stability parameter of the procedure. The construction of TV stable procedures is very similar to the construction of DP procedures (we elaborate on this relation in Appendix A). However, the way we use TV stable procedures is via the maximal coupling relation which gives that the probability of observing the same output under the original and updated dataset, is large. Reflection coupling is (one of the many) analytical techniques to explicitly construct couplings. In our setting, if the TV stability parameter is small (large noise), a rejection sampling step will likely result in acceptance and the original output is sufficient (i.e., no change after retraining). If it fails, then reflection, which produces a sample from the distribution under the updated set.
> We think that this “black-box” understanding of reflection coupling (and not how it achieves this, which is technical) should be sufficient to understand our method.
>
> **..hard to understand it at a conceptual level. Why and how does it perform unlearning?**
>
> Contrary to the reveiwer's claims, we explain the unlearning algorithm in text in the first paragraph of Section 4.2. Admittedly, there are details in the full algorithm that are not mentioned, but those are rather mechanical and unimportant, such as updating the internal state of nodes in the tree data structure. The key idea is to consider the vertices where the deleted points participate and in the order in which it participates, which is from the leaf to the root. Then starting from leaf, we start adjusting the non-noised output by subtracting the gradient of the deleted point and adding the gradient of the new point. Then we do rejection sampling, which ascertains if the original noisy sample still stuffices, if not we reflect. We also describe this in Figure 1.
>
> **In what sense is “linear queries” (Definition 8) is “linear in the data-points**
>
> They are linear in the usual sense we use when we define linear (vector) spaces of functions - think about queries as functions indexed by data-points. To remind, a vector space of functions uses component-wise addition (i.e. $f+g$ being $x\mapsto f(x)+g(x)$) as the “addition” operation and ($c.f$ being $x\mapsto c.f(x)$, where $c$ is in some field, say $\mathbb{R}$) as the “multiplication” operation. With these, it is easy to check that it (by definition) satisfies all criterias of linearity.

---

> > ### Comment · Reviewer_qNGF · 2022-11-17
> > **Thanks**
> >
> > Thank you for the response! The recent revision significantly improves the presentation of this paper.
> >
> > The part from the rebuttal on why adding noise is necessary should be added somewhere in the paper.
> >
> > I still find the algorithm description in section 4.2 quite unclear.
> >
> > 1. First paragraph of section 4.2, "let s be its binary representation [...]", I think it suffice to just say "let v_s be the leaf node that contains z_j."
> > 2. "try to replace node $v_s$ using $v_l$." --- from the figure it seems that $v_l$ is eventually deleted? Does the pseudocode do that as well? (Also the notation $\ell$ v.s. $l$ is inconsistent with the pseudocode (Algorithm 3).)
> > 3. "This amounts to making only two new queries, under the data point zj and the last point, with responses g and g′". This seems quite vague to me. First, g and g' should be expanded out, as in the pseudocode.  Second, based on the pseudocode, by "last point", you really mean the data point contained in the last leaf node.
> > 4. Why the rejection sampling in line 13? What's the target distribution here?

---

> > > ### Author Response · Authors · 2022-11-18
> > > **Response to further comments**
> > >
> > > We thank the reviewer for going through the revision and their follow-up comments which we think has helped improve the presentation of the paper. We have expanded the text in Section 4.2 (text in red in revision) which gives significantly more details to the unlearning process than before – we think that this should answer the reviewer’s questions. Below we respond to specific comments.
> > >
> > > **First paragraph of section 4.2, "let s be its binary representation [...]", I think it suffice to just say "let v_s be the leaf node that contains z_j."**
> > >
> > > We have made the modification suggested by the reviewer.
> > >
> > > **.. is eventually deleted? Does the pseudocode do that as well?**
> > >
> > > Yes, in the pseudo-code, in line 6, we indeed remove the last node from the tree. We have added it to the text in Section 4.2 as well.
> > >
> > > **.. notation is inconsistent with the pseudocode**
> > >
> > > Thanks for pointing this out; we have corrected this now.
> > >
> > > **First, g and g' should be expanded out, as in the pseudocode.**
> > >
> > > As suggested, we have added this to the text in Section 4.2.
> > >
> > > **.. you really mean the data point contained in the last leaf node**
> > >
> > > Correct, we have revised the text which hopefully clarifies this better.
> > >
> > > **Why the rejection sampling in line 13? What's the target distribution here?**
> > >
> > > The target distribution in rejection sampling is $\mathcal{N}(u’,\sigma^2\mathbb{I})$. To answer the "why  the rejection sampling" question, we explain the logic to the unlearning process below – we have added this discussion (with more details) in the revision.
> > >
> > > During unlearning, we simulate (roughly speaking) the dynamics of the learning algorithm if the deleted point was not present to begin with. In that case, in place of the deleted point $z$, some other point would have been used there. Since the dataset was randomly permuted, every point is equally likely to be used, and so we can simply use the last point $z'$ in the tree, in its place (this choice of last point is important for computational efficiency). Then, we need to replace all the computations which used the deleted point $z$ with the same computation under $z'$. Starting with leaf node $v_s$, we update the original unperturbed prefix-sum query response under $z$ i.e.  $u$  to what it would have been under data-point $z'$: $u' = u-g'+g$. Further, since the training method adds noise $\mathcal{N}(0,\sigma^2\mathbb{I})$ to $u$ to produce original noisy response $r$, we now need to produce a sample from $\mathcal{N}(u',\sigma^2\mathbb{I})$ to satisfy exact unlearning. Naively, we could simply get a *fresh* independent sample from $\mathcal{N}(u',\sigma^2\mathbb{I})$, however, this would change the noisy response $r$, and hence require all subsequent computations to be redone (the adaptive nature of the process). So, ideally, we want to reuse the same $r$ and yet generate a sample from $\mathcal{N}(u',\sigma^2\mathbb{I})$. This is precisely the problem of constructing a maximal coupling we discussed in the preliminaries, wherein we also discussed one method called reflection coupling to do. The reflection coupling method is based on a rejection sampling procedure which (roughly speaking) ascertains if the original response $r$ is still sufficient under the new distribution $\mathcal{N}(u',\sigma^2\mathbb{I})$. If it results in acceptance (likely to happen for TV stable procedures), then we move to the parent of the node and repeat, if some step fails, we do the reflection step which changes the noisy response and then retrain from the next leaf in the tree.
> > >
> > > We hope this clarifies the questions pertaining to the unlearning procedure. We also encourage the reviewer to go through the text in red in Section 4.2 in the revision. If there are any more questions/suggestions, we are happy to address them.

---

> > > > ### Comment · Reviewer_qNGF · 2022-11-21
> > > > **Re**
> > > >
> > > > Thank you for the revision! I think much of my early concerns have been addressed and so I raised my rating.

---

> > > > > ### Author Response · Authors · 2022-11-21
> > > > > **Thank you**
> > > > >
> > > > > We thank the reviewer for raising the score, and also for their time and effort towards reading the paper. We acknowledge that their suggestions have helped us improve the paper.

---

> ### Author Response · Authors · 2022-11-11
> **Response to Reviwer qNGF (part 2)**
>
> **After Definition 6 it should be mentioned that the bounded sensitivity condition is satisfied in a variety of applications.**
>
> Thanks, we have added this in the revision.
>
> **What does Algorithm 6 do at an intuitive level? What is the LearnLinearQueries function?.. suggest the author(s) expand this section (Appendix B) and put it into the main paper as a warm up before delving into the prefix sum queries**
>
> We thank the reviewer for going through this appendix section and their suggestions - we have made the corrections pointed out, and added details. We did not include this part owing to space constraints, and that most of the non-trivial work is required for the prefix-sum query part.
>
> About Algorithm 6, this is very similar to the prefix-sum query unlearning algorithm, however the iterations here are arranged as a chain (which is more standard) rather than a tree. So, it starts from the first iteration, performs rejection sampling, if it results in acceptance, then proceeds to the second iteration and so on. If some iteration results in rejection, then it does the reflection step, and continues retraining from there on – the LearnLinerQueries function is simply the learning algorithm (Algorithm 5), we have corrected this typo in the revision.
>
> **... there’s been a flurry of recent work on machine unlearning. This includes, fThis includes, for example, Descent-to-Delete: Gradient-Based Methods for Machine Unlearning by Seth Neel, Aaron Roth, Saeed Sharifi-Malvajerdi (ALT 21)**
>
> We indeed cite this paper in the related work section (among other recent works in this area) but the reason we don’t compare is due to the difference between the unlearning criterions (exact vs approximate unlearning), which unfortunately prevents a fair comparison (strictly speaking). We have revised the discussion in the revision.

---

### Official Review · Reviewer_NHVJ · 2022-10-23

**Confidence:** 2
**Correctness:** 4
**Technical Novelty And Significance:** 2
**Empirical Novelty And Significance:** Not applicable
**Recommendation:** 6

**Clarity, Quality, Novelty And Reproducibility:**

The work is fairly well-written (with exceptions observed above), and has reasonable quality applications. The framework introduced for adaptive query algorithms is original, though it draws heavily on analysis techniques and algorithms from prior work.

**Strength And Weaknesses:**

Strengths:

1. The authors introduce the first general framework for (exact) machine unlearning for adaptive algorithms using linear and prefix query algorithms
2. They give improved minimax rates for unlearning in stochastic convex optimization, a ubiquitous paradigm.
3. They give many applications (federated learning, k-means, various SCO settings…), and generalize to streaming setting
4. The paper is generally well written (with some exceptions detailed below)


Weaknesses

1. While machine unlearning is motivated by data privacy/rights as a general concept, the exact model studied in this paper is not clearly well motivated. Approximate machine unlearning (or any say computationally indistinguishable method) seems substantially more useful. Furthermore, while the authors claim that GDPR and CCPA require companies to fully retrain models in this sense, as far as I can tell these laws say nothing of the sort, and only include vague instructions that personal data should be removed upon request. I find the introductory paragraph to be very misleading in this sense. This said, there is legal precedence for requiring model re-training in a 2021 Federal Trade Commission decision, but this is a specific instance and should be cited as such. It seems very plausible that approximate unlearning would be accepted in general.
2. Many of the results seem to be straightforward generalizations or combinations of prior work and techniques, though proving the prefix query theorem does require some involved technical calculations. Furthermore, since no lower bounds are given in the paper it is hard to tell how substantial the rate improvements are.
3. A more minor complaint is that the notion of relative unlearning complexity needs better motivation. Of course the idea of algorithms whose unlearning time is faster than retraining is natural, but it seems like the denominator comparison should maybe be to the best possible rate. For instance, one could have a very poor unlearning algorithm with very good relative complexity, where it would be better to just run a different method from scratch.
4. No accuracy guarantee is given for the general framework, and has to be shown on a by-algorithm basis. The authors state informally that accuracy should be preserved for noise-tolerant algorithms, but if one is giving a general framework for application this should be formalized.
5. While the level of detail given in the main body is ok for the short version, there needs to be additional background and exposition somewhere in the paper. It should not be necessary to be familiar with any specific prior work to understand the paper. I'd also recommend defining even very basic terms (at least briefly) such as query release, for non-familiar readers.

Typos:

Page 5: In 2. Efficient Unlearning Model should “does” be “doesn’t”?

Page 7: Line 11 does not seem to be rejection sampling. Perhaps line 13?

**Summary Of The Paper:**

The authors propose a general framework for machine unlearning via (structured) adaptive algorithms. In machine unlearning, given a model trained via dataset S, one attempts to build algorithms which can efficiently remove any point z from the dataset post-training in such a way that the altered model is indistinguishable from training on $S \setminus \{z\}$. The main goal is to build algorithms where the cost of unlearning is significantly less than the cost of simply retraining the model. The study of machine unlearning is practically motivated by data privacy laws and `the right to be forgotten,’ in particular the General Data Protection Regulation (GDPR), and California Consumer Privacy Act (CCPA), which require (in limited circumstances) that a company must delete stored personal data on request.

In this work, the authors generalize the prior techniques of Ullah et al. (COLT 2021) for machine unlearning via (noisy) SGD via total variation stability to hold for a general class of adaptive algorithms built on either "linear" or "prefix-sum" queries, two well-studied classes of algorithms within optimization and privacy. In particular, they show how to modify any such algorithm into one in which unlearning costs only a $\rho$-fraction as many queries as retraining (for arbitrary $\rho > 0$). The main contribution is the algorithm and analysis for prefix queries, which is roughly based on combining Ullah et al.’s TV-stability techniques with the binary tree mechanism of Dwork, Naor, Pitassi, and Rothblum (STOC 2010). The authors also study an extension of these methods to the streaming setting, where multiple insertions and deletions in such a manner in an online fashion.

Finally, authors give a number of applications of this general framework based on known algorithms in the literature, namely for stochastic convex optimization (SCO) via Frank-Wolfe and Dual Averaging, and dimension-independent bounds for GLMs via the above and Johnson-Lindenstrauss (as in Arora et al. 2022). These techniques provide the best known unlearning algorithms for SCO, improving the known convergence rate from $\frac{1}{\sqrt{n}} + (\frac{\sqrt{d}}{n\rho})^{2/3}$ to $\frac{1}{\sqrt{n}} + (\frac{\sqrt{d}}{n\rho})$, thus showing unlearning has no asymptotic cost up to relative complexity $\sqrt{d/n}$. The authors also give applications of the linear query method to federated learning and k-means.

**Summary Of The Review:**

This is a nice follow up work to Ullah et al. 2021, the idea of building a general framework of machine unlearning for adaptive query algorithms is a good idea, and the authors give a number of applications. On the other hand, it is not clear that the model of unlearning (and complexity measure) the authors study is interesting or well-motivated, and at a technical level it is also not clear much additional insight is given over prior works. As it stands, I am not sure this work is quite above the bar of ICLR.

Edits: See rebuttal and response, raising to above bar.

---

> ### Author Response · Authors · 2022-11-10
> **Response to Reviewer NHVJ (part 1)**
>
> We thank the reviewer for carefully reading the paper and writing a detailed review. We acknowledge that the reviewer makes some very interesting points and we believe that our response should adequately answer these. If not, we are happy to engage in further discussions. We also note that we have revised the paper to incorporate their comments (see text in red in the revision).
>
> **...find the introductory paragraph to be misleading in this sense... Approximate machine unlearning (or any say computationally indistinguishable method) seems substantially more useful.. seems very plausible that approximate unlearning would be accepted in general**
>
> This is a very good question. As suggested, we have revised the first paragraph (see text in red in the revision). There are mainly two reasons for the exact unlearning model:
>
> (a). *What is the right notion of approximation in approximate unlearning*: Firstly, it is not a-priori clear which notion of approximation in approximate unlearning criterion is appropriate. There are a number of works on approximate unlearning and there are differences between their notions. Moreover, as the reviewer themselves points, there is legal precedence on requiring retraining (or exact unlearning). This demonstrates that our definition is indeed natural and useful.
>
> (b). *Understanding the limits of exact unlearning could better motivate approximations*: Our second point is more pedagogical; we think that understanding the limits of what is possible in the exact unlearning model should precede any relaxations. This is consistent with how scientific progress has been made across a multitude of disciplines and/or how they are taught – a course in approximation algorithms typically looks at problems (believed) hard to solve exactly (eg: MAX-CUT),  approximations in optimization are motivated by the futility of an exact worst-case model, in cryptography, statistical indistinguishability (and its limits) motivates computational indistinguishability, and there are many other examples.
>
> In the problem of unlearning, we seem to want to jump straight to approximate notions – arguably, this is due to the rapid development of the area of differential privacy and the incentive to make the problem look similar to facilitate (easy) transfer of tools.  However, note that even in differential privacy itself, the approximation is motivated by results which demonstrate impossibility of any non-trivial utility with the exact model (see Dwork and Naor, 2008, Section 2). Similarly, we think that pursuing the limits of what is possible in the exact model will set forth concrete motivation for what else is to be gained with approximations.
>
> Dwork and Naor, 2008: On the Difficulties of Disclosure Prevention in Statistical Databases or The Case for Differential Privacy.
>
> We also note that the reviewer perhaps incorrectly conflates (typical) approximate unlearning with computational indistinguishability. The most common approximate notion is based on differential privacy which is based on statistical indistinguishability, and not computational indistinguishability which is commonplace in cryptography. Extending the reviewer's logic, even for differential privacy, a computational definition should be sufficient for all practical purposes, and hence we should abandon the current definition. However, the wide-spread adoption of differential privacy demonstrates that this is not the case, since it turns out that (arguably) this pessimistic model allows design of simple and powerful procedures which are still good enough for practical purposes.
>
> To conclude about the usefulness of our defintion, we opine that the usefulness in real world applications can only be judged a-posteriori, which will take some time. Our exact unlearning model is sufficiently natural enough to merit scientific investigation (also, note the legal precedent that the reviewer points to).
>
>
>  **the notion of relative unlearning complexity needs better motivation**
>
> This is a good point. Indeed, relative unlearning complexity in itself does not completely capture if the unlearning procedure is good or not. We never claimed that it, in itself, is a complete and/or sound measure of unlearning efficiency (it is not). However, note that in our applications, all our learning methods are linear-time, so the denominator is as small as it can be (upto-constants). We have added these clarifications to the text (see text in red in the revision).

---

> ### Author Response · Authors · 2022-11-10
> **Response to Reviewer NHVJ (part 2)**
>
> **..since no lower bounds are given in the paper..**
>
> Can the reviewer clarify what is the setting and the quantity on which the reviewer is inquiring about lower bounds? In the current setup, there are many resources: accuracy, learning runtime, unlearning runtime, in addition to the unlearning procedure satisfying correctness. One formulation could be the smallest unlearning runtime among all correct, linear time and optimal rate procedures. However, we think that this may require significant work. We note that even with DP (arguably a simpler restriction on algorithms than satisfying unlearning correctness), there are hardly any (non-trivial) lower bounds in terms of oracle/information complexity for optimization.
>
> **results seem to be straightforward generalizations or combinations of prior work and techniques**
>
> We stress that our results are in no way straightforward generalizations of prior work. As the reviewer has noted, the main result of the prefix sum query is non-trivial. Further, even in applications, we make new contributions. For example: as a corollary, our result in the smooth case gives a method for DP smooth SCO with optimal rate in the setting where where you need to release a model after observing one data point.
> To elaborate, there were three known linear-time methods for this problem which provide optimal rates:
> (a). Snowball SGD [Feldman, Koren and Talwar, STOC 20] - which releases only the last iterate
> (b). Phased SGD [Feldman, Koren and Talwar, STOC 20] - which releases $\log{n}$ iterates,
> (c). Dyadic mini-batched Frank-Wolfe method [Bassily, Guzman and Nandi, COLT 21] – a recent work which uses a variant of tree-aggregation which requires growing multiple trees of varying depth, together with mini-batching; this also does not release all iterates.
>
> In contrast, our method, based on a much simpler tree aggregation, allows releasing iterates after every iteration processing one data point each. In the context of unlearning, this, besides providing a unified presentation for the smooth and non-smooth case, is integral for the correctness of our unlearning procedure. Furthermore, this also allows us to generalize to the setting of dynamic streams.
> We do not stress on this, partly due to space constraints, and to keep the paper more focused on the unlearning problem.
>
> **no accuracy guarantee is given for the general framework**
>
> While it is easy to give accuracy guarantees in terms of total prefix-sum query error (as in DP continual release results), we do not do it since it does not apply directly to our applications. The analysis for the smooth and non-smooth case requires analysis which is sufficiently tailored to the respective settings.  We do not think this is a major limitation.

---

> > ### Comment · Reviewer_NHVJ · 2022-11-19
> > **Reviewer Response**
> >
> > We thank the authors for their in depth response, especially re: motivation for exact unlearning and in further clarifying the novelty of their method and applications. The added exposition in the new version is also very helpful. With this in mind I have raised the review score.
> >
> > The lower bound statement was meant with respect to rate bounds on population risk ($d$, $n$, and $\rho$), but as the authors suggest it could be such bounds require substantial effort/new tools. Such bounds would be helpful towards understanding the progress this work makes towards optimality and limitations of the exact model.

---

> > > ### Author Response · Authors · 2022-11-20
> > > **Thank you**
> > >
> > > We thank the reviewer for going through our response and revision and increasing their score.

---

### Official Review · Reviewer_VmwR · 2022-10-25

**Confidence:** 2
**Correctness:** 3
**Technical Novelty And Significance:** 3
**Empirical Novelty And Significance:** Not applicable
**Recommendation:** 5

**Clarity, Quality, Novelty And Reproducibility:**

Clarity: The paper is difficult to read as it is heavy on content and notation. I was not sure why noise needed to be added. It seems it is not for DP purposes. Is it to make learning fall into distribution that would be close to unlearning? Though I appreciate the tree and the figure, it did not make things much clearer. It would be best to define all parameters and use fonts that fit.

Quality: The authors use unlearning definition which is not quite what one would call exact unlearning: exact unlearning is the model one obtains if the data point is deleted and retraining happens. It seems definition 1 is a probabilistic guarantee. That is learning process is made probabilistic so that unlearning of probabilistic nature would be an appropriate counterpart. Please consider definitions in Thudi et al. USENIX Security 2022.

Novelty: The idea of decomposing some training algorithms into prefix-sum is nice. It would be good to compare with Bourtoule et al as sharding seem to also isolate points and recompute those shards where the deleted point was used. Though authors say this work does not provide theoretical guarantees it seems to not use a data point at all hence, may provide exact unlearning as well.

Reproducibility: N/A as this is a theoretical paper.

**Strength And Weaknesses:**

Strengths:
- the problem of unlearning is really interesting and important
- the authors aim to propose a method that is more efficient than training from scratch
- the idea of decomposing into prefix-sum-like learning is nice

Weaknesses:
- the definition of exact unlearning used by authors is not common
- the paper is extremely dense in notation and content
- the method seems to work only for unlearning one point though the authors allude it may be extended
- the method is expensive in terms of space as it seems one need store a model for every data point
- no experimental results

**Summary Of The Paper:**

The authors consider the problem of unlearning. The aim is to do more efficiently than retraining from scratch. The authors propose to construct a prefix-sum-like tree during learning/training to keep intermediate results. These results can then be queried to construct a model that will look like the one without a certain data point (i.e., the one that was requested to be deleted).

**Summary Of The Review:**

I believe the paper is not ready to be published due to its presentation (the method description) and how it positions itself (i.e., is this exact unlearning, how it compares to retraining from scratch, how it compares to sharding approach by Bourtoule et al. It may be that in considers inly certain training tasks but it would be best to be clear about it.

---

> ### Author Response · Authors · 2022-11-10
> **Response to Reviewer VmwR (part 1)**
>
> We thank the reviewer for their feedback. We note that we have revised the paper to incorporate their comments (see text in red in the revision). We answer the reviewer’s questions below.
>
> **... I was not sure why noise needed to be added. It seems it is not for DP purposes. Is it to make learning fall into distribution that would be close to unlearning? ..**
>
> The problem of unlearning is concerned with the design of both the learning and (corresponding) unlearning algorithm for a specified task. In our work, we use randomized learning algorithms (eg: permuting the dataset and adding noise) since it facilitates efficient unlearning.
>
> About how noise helps: you are right that the noise is not for DP purposes – in our work, we are not concerned about constructing differentially private procedures. However, the noise (or randomization, in general) can also help in improving unlearning complexity. To give some intuition, consider some task, eg; mean computation, and think about the two extremes: (a) no noise, and (b) large (infinite) noise. In the first case, the output is most accurate, but requires (potentially) full retraining during unlearning. In the second case, the output has trivial accuracy, but requires no work during unlearning, since the output is just noise regardless of the dataset. Adding controlled noise allows one to trade-off accuracy with unlearning complexity.
>
> We model this intermediate point by the Total Variation (TV) stability parameter of the learning procedure. The construction of TV stable procedures is very similar to the construction of DP procedures (we elaborate on this relation in Appendix A). However, the way we use TV stable procedures is via the  maximal coupling relation which gives that the probability of observing the same output under the original and updated dataset, is large. Reflection coupling is (one of the many) analytical techniques to explicitly construct maximal couplings. In our setting, if the TV stability parameter is small (large noise), a rejection sampling step in reflection coupling will likely result in acceptance and the original output is sufficient (i.e., no change in model output upon unlearning). If it fails, then reflection, which produces a sample from the distribution under the updated set. Also, see the text in red in Section 2 in the revision.
>
> **The authors use unlearning definition which is not quite what one would call exact unlearning: exact unlearning is the model one obtains if the data point is deleted and retraining happens...**
>
> We are not sure we understand the point here. Speficically, what does the reviwer mean by *“exact unlearning is the model one obtains if the data point is deleted and retraining happens.”* If (re)training is a randomized method, then one would not obtain the same model even if the training algorithm is run twice on the same dataset. Also, we would like to remind the reviewer that most widely-used machine learning algorithms are indeed randomized – the S in SGD stands for Stochastic. Moreover, from an optimization perspective which is the dominant framework of algorithmic techniques for machine learning, randomization is necessary for fast learning algorithms  – see, for example, Woodworth and Srebro 2017, where they show a separation of complexity between deterministic and stochastic schemes.  In that case, an equality of distributions is indeed the correct generalization of equality. This is indeed what was proposed in the early work of Ginart et al 2019 on machine unlearning, which we use. We also thank the reviewer for sharing the reference; however, can the reviewer point to a specific definition in the paper they have in mind?
>
> Woodworth and Srebro, Tight complexity bounds for optimizing composite objectives, Neurips 2017
>
> A Ginart, M Guan, G Valiant, JY Zou, Making AI forget you: Data deletion in machine learning, Neurips 2019
>
> **Handling multiple requests**
>
> In Section F in the Appendix, we indeed present a generalization to the streaming setting consisting of insertions and deletions. Similarly, our schemes can be modified in standard ways to handle batch deletion requests.

---

> ### Author Response · Authors · 2022-11-10
> **Response to Reviewer VmwR (part 2)**
>
> **..dense in notation and content**
>
> Our work is theoretical in nature, so some of the notation is unavoidable so as to be precise. We have since revised the paper (see text in red) which hopefully provides more intiution behind the techniques.
>
> **It would be good to compare with Bourtoule et al... Though authors say this work does not provide theoretical guarantees it seems to not use a data point at all hence, may provide exact unlearning as well.**
>
> Yes, the method proposed in Bourtoule et a. satisfies exact unlearning – we say this in the paper - “The work of Bourtoule et al. (2021) proposes a general methodology for exact unlearning..” However, it provides no guarantees on *accuracy*, even in simple convex settings – we have since revised the sentence to add this clarification. Without an accuracy guarantee, a trivial algorithm evaluating a constant function also satisfies exact unlearning. Similarly, our general framework of adaptive query release is meaningless if not the applications yield non-trivial results. We acknowledge that the goal in Bourtoule et al. is to give practical heuristics, whereas, we want to understand the complexity of unlearning, hence an accuracy guarantee is required in our case so as to eliminate trivialities.
>
> **the method is expensive in terms of space.. no experimental results..**
>
> Our work is theoretical in nature wherein the main goal is to improve the unlearning complexity without heed to constraints such as space. This is exactly the framework in which our current understanding of (information) complexity of (convex) optimization rests.

---

> ### Author Response · Authors · 2022-11-21
> **Concerns?**
>
> We hope that the reviewer lets us know of any pending concerns which our respose has not addressed. We will be happy to engage in further discussion towards resolving them.

---

### Decision · Program_Chairs · 2023-01-20

**Decision:**

Reject

**Justification For Why Not Higher Score:**

The exposition could be improved to make it more accessible to the broader ICLR community.

**Justification For Why Not Lower Score:**

N/A

**Metareview: Summary, Strengths And Weaknesses:**

The paper studies the problem of machine unlearning where the goal is to update a model without retraining from scratch in response to deletions of training datapoints. The paper considers exact unlearning where the updated model is indistinguishable from the model obtained by retraining from scratch. The paper designs unlearning algorithms that improve the excess population risk for linear and prefix-sum queries, and they apply to several settings of interest, including unlearning in the context of both smooth and non-smooth stochastic optimization and generalized linear models.

The reviewers appreciated the theoretical contributions of the paper. In particular, the algorithm and analysis for prefix sum queries introduces novel ideas and techniques that were appreciated by the reviewers. The reviewers were concerned about the paper's writing and its technical clarity and accessibility to a broad audience. The authors revised the exposition based on the reviewers' feedback. During the subsequent discussion, the reviewers noted that the revision improved the exposition but the paper could further benefit from a more substantial revision that will make it more accessible to the broader ICLR community. Some specific suggestions are the following:

- Provide an overview of the pseudocode in the main body.
- Explaining in the main body the algorithm and results on a concrete application will facilitate understanding. One of the applications discussed in the appendix, the kmeans or federated learning, could be brought forward and used for this purpose.
- Illustrate how the guarantees translate to specific examples of interest.
- Consider some experiments on synthetic data as a proof of concept for the algorithms.
- Add an analysis of both space and time, discuss that the entire model needs to be stored.

We encourage the authors to further revise the paper to make it more accessible to a broader machine learning audience.

**Summary Of Ac-Reviewer Meeting:**

We discussed the theoretical contributions of the paper, which the reviewers appreciated. We also discussed the exposition of the paper. The reviewers felt that the exposition remained very dense and not very accessible to a broad machine learning audience. This is in part due to the fact that it is a theoretical paper that requires a fair amount of background in order to understand the algorithm and the analysis. Nevertheless, the reviewers felt that the authors could improve the exposition substantially by incorporating some of the suggestions outlined above.